# Termination of non-coding transcription in yeast relies on both an RNA Pol II CTD interaction domain and a CTD-mimicking region in Sen1

Zhong Han[1,2,†], Olga Jasnovidova[3,‡], Nouhou Haidara[1,2], Agnieszka Tudek[1,#], Karel Kubicek[3], Domenico Libri[1] (iD), Richard Stefl[3] (iD) & Odil Porrua[1,*] (iD)

## Abstract

**Pervasive transcription is a widespread phenomenon leading to the production of a plethora of non-coding RNAs (ncRNAs) without apparent function. Pervasive transcription poses a threat to proper gene expression that needs to be controlled. In yeast, the highly conserved helicase Sen1 restricts pervasive transcription by inducing termination of non-coding transcription. However, the mechanisms underlying the specific function of Sen1 at ncRNAs are poorly understood. Here, we identify a motif in an intrinsically disordered region of Sen1 that mimics the phosphorylated carboxy-terminal domain (CTD) of RNA polymerase II, and structurally characterize its recognition by the CTD-interacting domain of Nrd1, an RNA-binding protein that binds specific sequences in ncRNAs. In addition, we show that Sen1-dependent termination strictly requires CTD recognition by the N-terminal domain of Sen1. We provide evidence that the Sen1-CTD interaction does not promote initial Sen1 recruitment, but rather enhances Sen1 capacity to induce the release of paused RNAPII from the DNA. Our results shed light on the network of protein–protein interactions that control termination of non-coding transcription by Sen1.**

**Keywords** non-coding transcription; pervasive transcription; RNA polymerase II CTD; Sen1 helicase; transcription termination
**Subject Categories** Chromatin, Transcription & Genomics; RNA Biology
**The EMBO Journal (2020) 39: e101548**

## Introduction

The concept of pervasive transcription emerged over a decade ago upon the discovery that a large fraction of both the prokaryotic and the eukaryotic transcriptomes is composed of non-coding RNAs (ncRNAs) without any obvious function. Pervasive transcription is potentially harmful for cell homeostasis since it can interfere with normal transcription of canonical genes and provoke the accumulation of toxic RNAs. Therefore, all organisms studied to date have evolved different mechanisms to circumvent the negative consequences of pervasive transcription. These mechanisms often rely on transcription termination and RNA degradation (for review, see Jensen *et al*, 2013).

In *Saccharomyces cerevisiae*, there are two major pathways for termination of RNA polymerase II (RNAPII) transcription. A pathway that depends on a macromolecular complex including the cleavage and polyadenylation factor (CPF) is essentially responsible for transcription termination at protein-coding genes, whereas the Nrd1-Nab3-Sen1 (NNS) complex targets a large fraction of the non-coding RNAs (ncRNAs) produced in the cell. Specifically, the NNS complex terminates transcription of most snoRNAs and a class of ncRNAs dubbed CUTs, for cryptic unstable transcripts, that constitutes the major product of pervasive transcription (for review, see Porrua & Libri, 2015). While snoRNAs are important for the correct modification of rRNA, CUTs are generally considered as non-functional molecules (Wyers *et al*, 2005; Arigo *et al*, 2006; Thiebaut *et al*, 2006; Schulz *et al*, 2013).

Each pathway is associated with distinct nuclease and polyA-polymerase activities that determine the stability and functionality of the RNAs they take care of. Precursors of mRNAs are cleaved at their 3′ ends at the so-called polyA site and polyadenylated by Pap1, which stimulates subsequent export to the cytoplasm and translation (for review, see Porrua & Libri, 2015). In contrast, ncRNAs terminated by the NNS-dependent pathway are polyadenylated by Trf4, a component of the TRAMP complex. These RNAs are then targeted by the nuclear form of the exosome bearing the Rrp6 exonuclease, which catalyses either 3′ end maturation, in the case of snoRNAs, or complete degradation, in the case of CUTs (LaCava

1  Université de Paris, CNRS, Institut Jacques Monod, Paris, France[¶]
2  Université Paris-Saclay, Yvette, France
3  CEITEC-Central European Institute of Technology, Masaryk University, Brno, Czechia,
   *Corresponding author. Tel: +33 (0) 157278035; E-mail: odil.porrua@ijm.fr
   [†]Present address: Francis Crick Institut, London, UK
   [‡]Present address: Max Planck Institute for Molecular Genetics, Berlin, Germany
   [#]Present address: Department of Biophysics, Institute of Biochemistry and Biophysics, Polish Academy of Sciences, Warsaw, Poland
   [¶]Correction added on 1 April 2020, after first online publication: This affiliation was corrected because the name of the university changed.

*et al*, 2005; Vanacova *et al*, 2005; Wyers *et al*, 2005). Given the very divergent fate of the RNAs terminated by each pathway, several mechanisms have evolved to ensure the specific action of the different protein complexes on the right targets. These mechanisms involve both protein–protein and nucleic acid–protein interactions. Some of the protein interactions that are crucial for coordinated and efficient transcription-related processes as termination and 3′ end processing are mediated by the C-terminal domain (CTD) of the largest subunit of RNAPII. The CTD is a large and flexible domain composed of 26 repeats of the heptapeptide YSPTSPS that undergoes dynamic phosphorylation throughout the transcription cycle (for review, see Harlen & Churchman, 2017a).

The CPF complex interacts preferentially with the S2P-form of the CTD, which is more prominent during middle-late elongation, via the CTD interaction domain (CID) of its associated factor Pcf11 (Lunde *et al*, 2010). In addition, the recognition of several AU-rich sequences, among which the polyA site, by other subunits, mediates the specific recruitment of the CPF complex to mRNAs (Xiang *et al*, 2014). Similarly, the CID of Nrd1 within the NNS complex interacts with the CTD phosphorylated at S5, which is a mark of early elongation (Vasiljeva *et al*, 2008; Kubicek *et al*, 2012), and both Nrd1 and Nab3 recognize specific sequence motifs that are enriched at the target ncRNAs (Hobor *et al*, 2011; Lunde *et al*, 2011; Wlotzka *et al*, 2011; Porrua *et al*, 2012; Schulz *et al*, 2013). Both the interaction with the CTD and with the nascent RNA contribute to the early recruitment of the Nrd1-Nab3 heterodimer and the efficiency of transcription termination (Gudipati *et al*, 2008; Vasiljeva *et al*, 2008; Tudek *et al*, 2014). The current model posits that Nrd1-Nab3 would recruit the helicase Sen1 that in turn promotes the final step in transcription termination (i.e. the release of RNAPII and the nascent RNA from the DNA, Porrua & Libri, 2013). Subsequently, a complex network of possibly redundant protein–protein interactions involving Nrd1, Nab3 and different components of the TRAMP and exosome complexes promotes efficient degradation of the released ncRNA (Tudek *et al*, 2014; Fasken *et al*, 2015; Kim *et al*, 2016).

As mentioned above, Sen1 is responsible for dissociation of the elongation complex. Sen1 is a highly conserved RNA and DNA helicase belonging to the superfamily 1 of helicases (Martin-Tumasz & Brow, 2015; Han *et al*, 2017). Transcription termination by Sen1 involves its translocation along the nascent RNA towards RNAPII and possibly subsequent contacts between specific regions of Sen1 helicase domain and the polymerase (Porrua & Libri, 2013; Han *et al*, 2017; Leonaitè *et al*, 2017). Neither *in vitro* nor *in vivo* Sen1 exhibits any sequence-specific RNA-binding capability (Creamer *et al*, 2011; Porrua & Libri, 2013); implying that Sen1 activity should be regulated in order to ensure its specific action on ncRNAs. Among the mechanisms that might contribute to keep Sen1 under control, we can cite: (i) the relatively low levels of Sen1 protein (63–498 molecules/cell depending on the study Ghaemmaghami *et al*, 2003; Newman *et al*, 2006; Kulak *et al*, 2014; Chong *et al*, 2015); (ii) its low processivity as a translocase, which makes termination highly dependent on Sen1 efficient recruitment to the elongation complex and RNAPII pausing (Han *et al*, 2017) and (iii) the interaction of Sen1 with the ncRNA targeting proteins Nrd1-Nab3, as evoked above.

Although it has been proposed that Nrd1 and Nab3 function as adaptors that provide the necessary specificity to Sen1, the precise regions involved in the interaction between these factors and whether these interactions are actually sufficient for timely Sen1

recruitment remains unclear. In this study, we identify and characterize the key interactions involved in Sen1 function. Sen1 is composed of a central helicase domain [amino acids (aa) 1,095–1,876] that is sufficient for transcription termination *in vitro* (Han *et al*, 2017; Leonaitè *et al*, 2017), together with a large N-terminal domain (aa 1–975) and a C-terminal intrinsically disordered region (1,930–2,231). Here, we show that the C-terminal end of Sen1 contains a short motif that mimics the phosphorylated CTD and is recognized by Nrd1 CID. We prove that this motif is the main determinant of the interaction between Sen1 and Nrd1-Nab3 heterodimer and provide the structural details of this interaction. Strikingly, we find that the CTD mimic in Sen1 is not a strict requirement for Sen1 function, although it contributes to Sen1 recruitment and to fully efficient termination at some targets. Instead, we show that the N-terminal domain of Sen1 promotes its interaction with the CTD of RNAPII and that this interaction is a global requirement for non-coding transcription termination under normal conditions. However, decreasing the transcription elongation rate renders the N-terminal domain less necessary for termination, which supports the notion that the Sen1-CTD interaction favours Sen1 action in the context of a kinetic competition between termination and elongation. We also find that the N-terminal and the C-terminal domains of Sen1 can interact with each other *in vitro*, which might modulate the interaction of Sen1 with RNAPII CTD and/or Nrd1. Our findings allow us to propose a detailed molecular model on how protein interactions can control the specific function of the transcription termination factor Sen1 on ncRNAs.

## Results

### Sen1 possesses a CTD mimic that is recognized by the CID domain of Nrd1

In a previous report (Tudek *et al*, 2014), we showed that Nrd1 CID domain can recognize a short sequence in Trf4 that mimic the S5P-CTD of RNAPII and that we dubbed NIM for Nrd1-Interaction Motif. A subsequent report described a second NIM in Mpp6, an exosome cofactor (Kim *et al*, 2016). During the course of our previous work, we discovered that the CID is also required for the interaction between Nrd1 and Sen1 (Fig 1A), an observation that was reported in an independent study (Heo *et al*, 2013). This prompted us to search for a putative NIM in Sen1 protein. The S5P-CTD and Trf4 NIM share three important features: (i) they contain one or several negatively charged aa at the N-terminal portion that interact with a positively charged surface of the CID; (ii) they contain a Y residue followed by several aa at the C-terminal part that adopt a β-turn conformation and interact with a hydrophobic pocket of the CID; and (iii) they are placed in protein regions that are predicted to be intrinsically disordered and therefore are fully accessible for the interaction with the CID. We identified a sequence in Sen1 C-terminal domain that fulfils the three characteristics and closely resembles the NIM in Trf4 (Fig 1B). Therefore, we tested the role of this motif by comparing the ability of wild type (wt) or ΔNIM versions of Sen1 to interact with Nrd1 by *in vivo* coimmunoprecipitation experiments using Nrd1-TAP as the bait (Fig 1C). Importantly, deletion of the putative NIM did not significantly alter the levels of Sen1 protein but dramatically reduced its interaction with Nrd1. Similar

experiments using Sen1 as the bait confirmed these results and showed that deletion of the NIM also strongly affects the association of Sen1 with Nab3 (Fig 1D). These results indicate that Sen1 NIM is the main determinant of the interaction of Sen1 with the Nrd1-Nab3 heterodimer. They also strongly suggest that Nab3 interacts with Sen1 via Nrd1.

The NIM is one of the very few sequence regions of the C-terminal domain of Sen1 that are conserved in the closest *S. cerevisiae* relatives, suggesting that this mode of interaction between Sen1 and Nrd1 is conserved in these yeast species (Fig EV1). Conversely, in agreement with previous data showing that Nrd1 and Sen1 orthologues do not interact with each other in *Schizosaccharomyces*

**A**

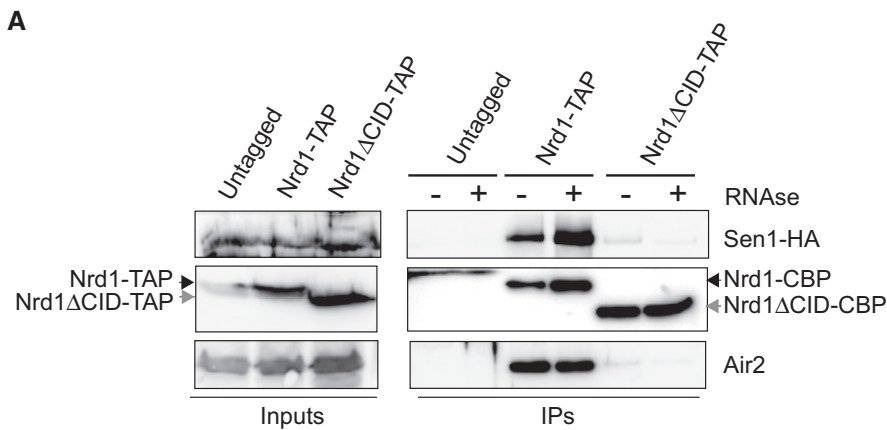

**B**

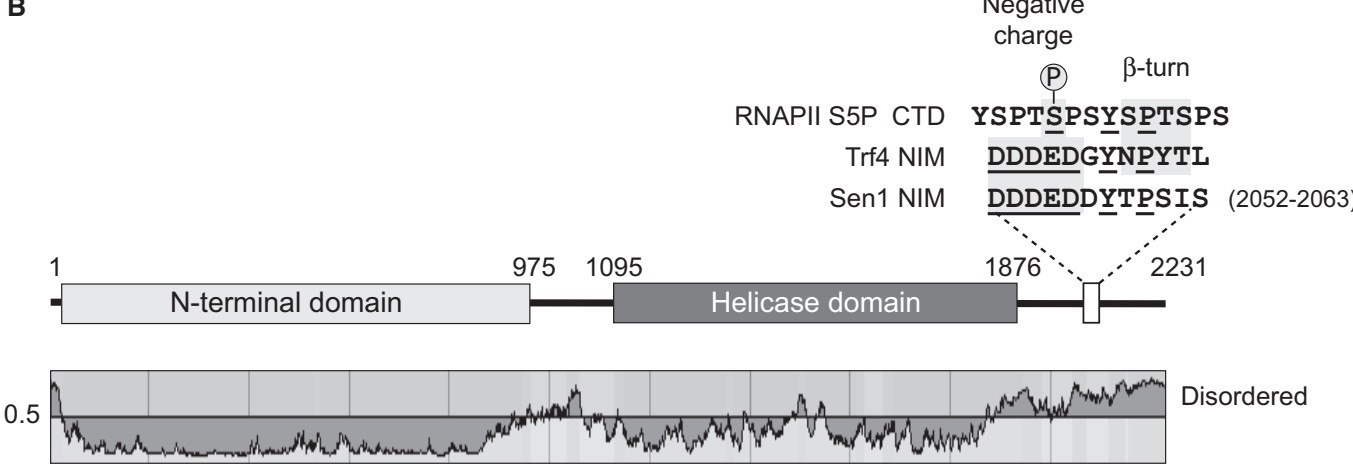

**C**

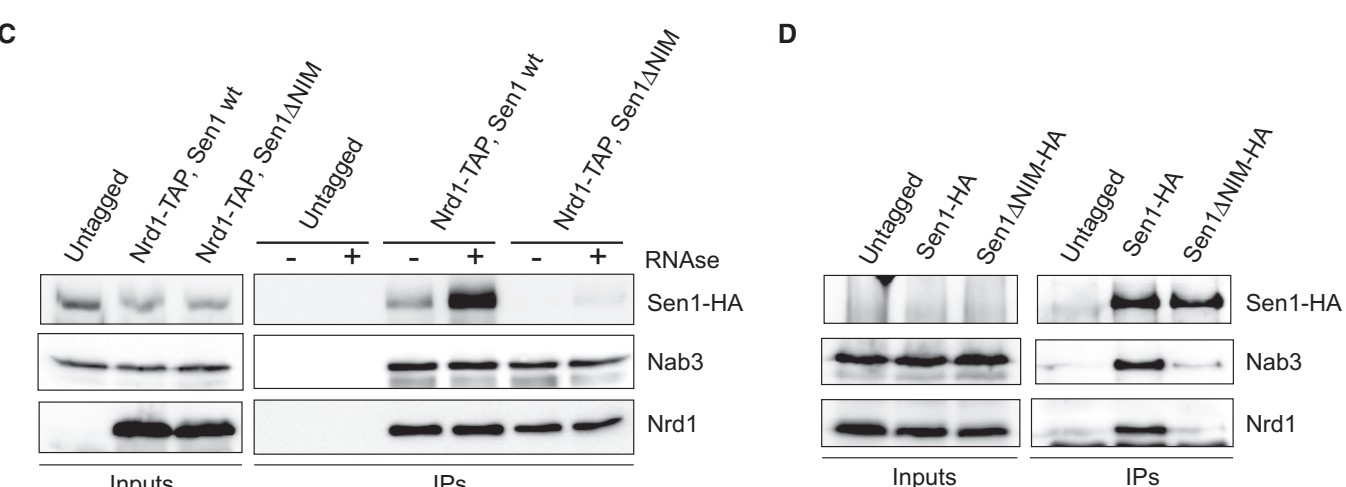

**Figure 1.**

◀

**Figure 1.  Identification of a Nrd1-Interaction Motif (NIM) in Sen1 that is critical for the integrity of the NNS complex.**

A   Deletion of the CID domain dramatically reduces the interaction of Nrd1 with Sen1. Coimmunoprecipitation (CoIP) experiments using TAP-tagged Nrd1 (either wt or ΔCID) as the bait. Representative gel of one out of two independent experiments.

B   Scheme of Sen1 protein. Globular domains are denoted by solid bars, whereas intrinsically disordered regions are shown by a line. The disorder prediction was obtained using IUPred (Dosztányi *et al*, 2005). The sequence of the RNAPII S5P-CTD and Trf4 and Sen1 NIMs is shown on the top. Structural elements that are important for the interaction with Nrd1 CID are indicated. Conserved positions are underlined.

C   Deletion of the NIM decreases substantially the association of Nrd1 with Sen1. CoIP experiments using Nrd1-TAP as the bait in a *SEN1* or *sen1ΔNIM* background. Representative gel of one out of two independent experiments.

D   CoIP experiments using HA-tagged Sen1, either wt or ΔNIM, as the bait. Representative gel of one out of two independent experiments. Protein extracts were treated with RNaseA prior to immunoprecipitation. In these experiments, Sen1 could not be detected in the input extracts.

Data information: Antibodies used for protein detection are listed in Appendix Table S3.

*pombe* (Lemay *et al*, 2016; Wittmann *et al*, 2017) and with the fact that no Nrd1 homologue could be identified in association with human Sen1(Yüce & West, 2013), we did not detect any putative NIM in Sen1 orthologues from these organisms.

**Structural analyses of the Nrd1 CID–Sen1 NIM interaction**

To compare the interaction of the newly identified Sen1 NIM (fragment harbouring aa 2,052–2,063) and the previously identified Trf4 NIM with Nrd1 CID, we performed a quantitative solution-binding assay using fluorescence anisotropy (FA) with purified recombinant Nrd1 CID and synthetic NIM peptides. We found that Nrd1 CID binds Sen1 NIM with a $K_D$ of 1.2 ± 0.02 μM, which is comparable to the dissociation constant of the Trf4 NIM–Nrd1 CID complex ($K_D$ 0.9 ± 0.02 μM), and roughly 100-fold smaller than the $K_D$ of the Nrd1 CID for the S5P-CTD (Tudek *et al*, 2014). In addition, we observed that the binding is similarly affected by previously described mutations in Nrd1 CID (L20D, K21D, S25D, R28D, M126A, I130K, R133A, see Appendix Table S1). The similar binding strength was also confirmed by [$^1$H,$^{15}$N] heteronuclear single quantum coherence (HSQC) titration experiment of Nrd1 CID. In this experiment, the protein amide resonances of Nrd1 CID were in slow or intermediate exchange regimes between their free and bound forms relative to the NMR timescale, when titrated with Sen1 NIM, as previously shown for the Trf4-NIM interaction (Tudek *et al*, 2014), which indicates that the interaction of Nrd1 CID with Sen1 NIM is quite stable. The analysis of changes in chemical shift perturbations (CSP) of Nrd1 CID in the presence of Sen1 NIM and Trf4 NIM peptides suggests that this domain employs the same interaction surface for both NIMs, with only a minor difference in the area of α4-helix and tip of α7-helix (Fig EV2).

In order to understand structural basis of the Nrd1–Sen1 interaction, we solved the solution structure of Nrd1 CID (1–153 aa) in complex with Sen1 NIM (Fig 2A; Appendix Table S2). The structure of Nrd1 CID consists of eight α helices in a right-handed superhelical arrangement as previously reported (Vasiljeva *et al*, 2008; Kubicek *et al*, 2012; Tudek *et al*, 2014). The Sen1 NIM peptide is accommodated in the binding pocket of Nrd1 CID in a similar manner to that of Trf4 NIM (Fig 2B and C). The upstream negatively charged part of the Sen1 NIM (D2052–D2053–D2054–E2055–D2056–D2057) interacts with a positively charged region of Nrd1 CID on tips of α1-α2 helices (Fig 2B). Charge-swapping mutations of in this region (K21D, S25D and S28D) resulted in a significant decrease in the binding affinity (Fig 2D; Appendix Table S1), confirming the importance of this region for the interaction. Furthermore, the L20D mutant also diminishes the binding affinity as it perturbs the overall geometry of the α1–α2 loop and thus the positioning of the

positively charged residues. The downstream hydrophobic part of Sen1 NIM (Y2058, T2059 and P2060) docks into a hydrophobic pocket of the CID formed by L127, M126 and I130. Y2058 makes a putative H-bond with D70 and R74 of the CID. Sen1 I2062 shows multiple intermolecular contacts in NMR spectra with the aliphatic groups of I130, R133 and S54 side chains. Furthermore, the neighbouring residue S2061 makes a putative H-bond with R74. As a result, these interactions induce the extended conformation of the downstream region of the Sen1 NIM, which contrasts with the formation of the canonical β-turn that was observed in the structures of CTD and NIM peptides bound to CIDs (Meinhart & Cramer, 2004; Becker *et al*, 2008; Lunde *et al*, 2010; Kubicek *et al*, 2012; Tudek *et al*, 2014). In these complexes, the peptides contain S/NPXX motifs that have a high propensity to form β-turns in which the peptides are locked upon binding to CIDs. In the case of Sen1 NIM, TPSI sequence is predicted as a non-β-turn motif (Singh *et al*, 2015). Indeed, we found that in the extended conformation the positioning of this downstream motif inside the hydrophobic area is energetically the most favourable. Our structural data suggest that Nrd1 CID is able to accommodate not only peptides with motifs that form β-turns but also peptides in the extended conformation that matches hydrophobicity and H-bonding partners inside the binding groove of the CID.

**A strong interaction between Sen1 and Nrd1-Nab3 is not essential for non-coding transcription termination**

Because of the importance of the NIM for the interaction between Sen1 and Nrd1-Nab3 heterodimer and its significant conservation among yeast species, we analysed the impact of the NIM deletion on growth and Sen1-mediated transcription termination *in vivo*. Surprisingly, we found that deletion of the NIM does not affect cell growth and only aggravated the thermosensitive phenotype of a Δ*rrp6* mutant, which lacks an exonuclease that plays a major role in degradation of ncRNAs targeted by the NNS complex (Fig EV3A).

In order to test for the role of Sen1 NIM in non-coding transcription termination, we performed RNAseq transcriptome analyses of Δ*rrp6* strains expressing either the wt or the ΔNIM version of Sen1. As expected, metagene analyses did not reveal any significant effect of the NIM deletion at protein-coding genes (Fig EV3B). Surprisingly, we did not observe any major difference in the expression profile of snoRNAs and CUTs in *sen1ΔNIM* (Fig EV3C and D), indicating that deletion of the NIM does not have a general impact on transcription termination at NNS targets. Because in our coimmunoprecipitation experiments, we observed that some minor interaction between Sen1 and Nrd1 persisted after the deletion of Sen1 NIM

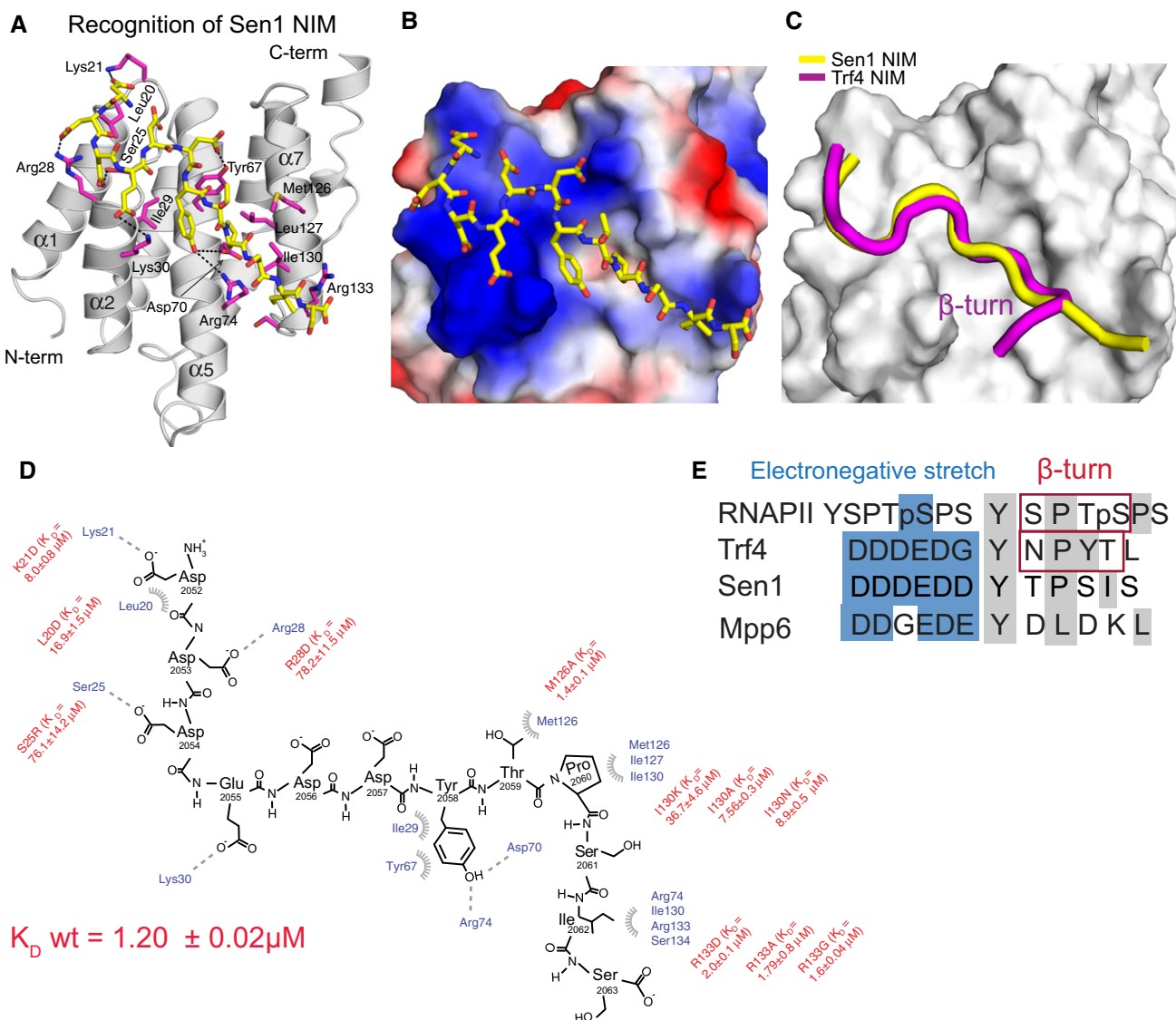

**Figure 2. Recognition of Sen1 NIM by Nrd1 CID.**

A NMR structure of Nrd1 CID bound to Sen1 NIM. The NIM peptide is represented in yellow sticks (only non-hydrogen atoms are shown), and Nrd1 CID is shown as a grey ribbon model. Nrd1 CID residues that form hydrophobic contacts and putative hydrogen bonds to Sen1 NIM peptide are shown in magenta sticks.

B Electrostatic surface representation of Nrd1 CID (electropositive in blue; electronegative in red; neutral in white) with the Sen1 NIM peptide (represented in yellow sticks; only non-hydrogen atoms are shown). The upstream electronegative stretch of Sen1 NIM interacts with an electropositive pocket of Nrd1 CID, while the C-terminal part of the peptide adopts an extended conformation that docks in a hydrophobic pocket of Nrd1 CID.

C Superposition of Nrd1 CID–Sen1 NIM (yellow) and Nrd1 CID–Trf4 NIM (magenta) complexes, displaying only peptide ribbons on the surface of Nrd1 CID. The comparison highlights the extended and β-turn conformations of Sen1 NIM and Trf4 NIM, respectively.

D Scheme showing contacts (putative H-bonds and hydrophobic contacts) and energetics between the Sen1 NIM peptide and Nrd1 CID. Equilibrium binding experiments with the protein mutants were monitored by FA. L20D mutant disrupts the hydrophobic contact with F17 and impairs the overall geometry of the α1-α2 loop that contributes to the interaction with the upstream electronegative stretch of NIM. $K_D$ (wild-type Nrd1 CID-Sen1 NIM) equals 1.20 ± 0.02 μM.

E Alignment of RNAPII CTD and CTD mimics (Trf4 NIM, Sen1 NIM and Mpp6 CTD mimic). Blue and grey boxes highlight the upstream electronegative stretches and hydrophobic regions, respectively. Previous structural works reported that S/NPXX motifs form the β-turn conformation (Kubicek *et al*, 2012; Tudek *et al*, 2014).

(Fig 1C); we considered the possibility that this remaining interaction could support sufficient levels of Sen1 recruitment, therefore explaining the weak phenotype of the *sen1ΔNIM* mutant. To investigate this possibility, we deleted the whole C-terminal domain (Cter) of Sen1 downstream of the previously identified nuclear localization signal (Nedea *et al*, 2008) and analysed the capacity of this mutant (Sen1ΔCter) to interact with Nrd1. Indeed, deletion of the Sen1 Cter

did not decrease the protein expression levels but abolished the interaction between Sen1 and Nrd1, indicating that this domain of Sen1 possesses additional surfaces that weakly contribute to the interaction between Sen1 and its partners (Fig EV3E). While this work was under revision, an independent study reported a secondary NIM within the Cter that mediates weak interactions with Nrd1 *in vitro*, further confirming our observations (Zhang *et al*,

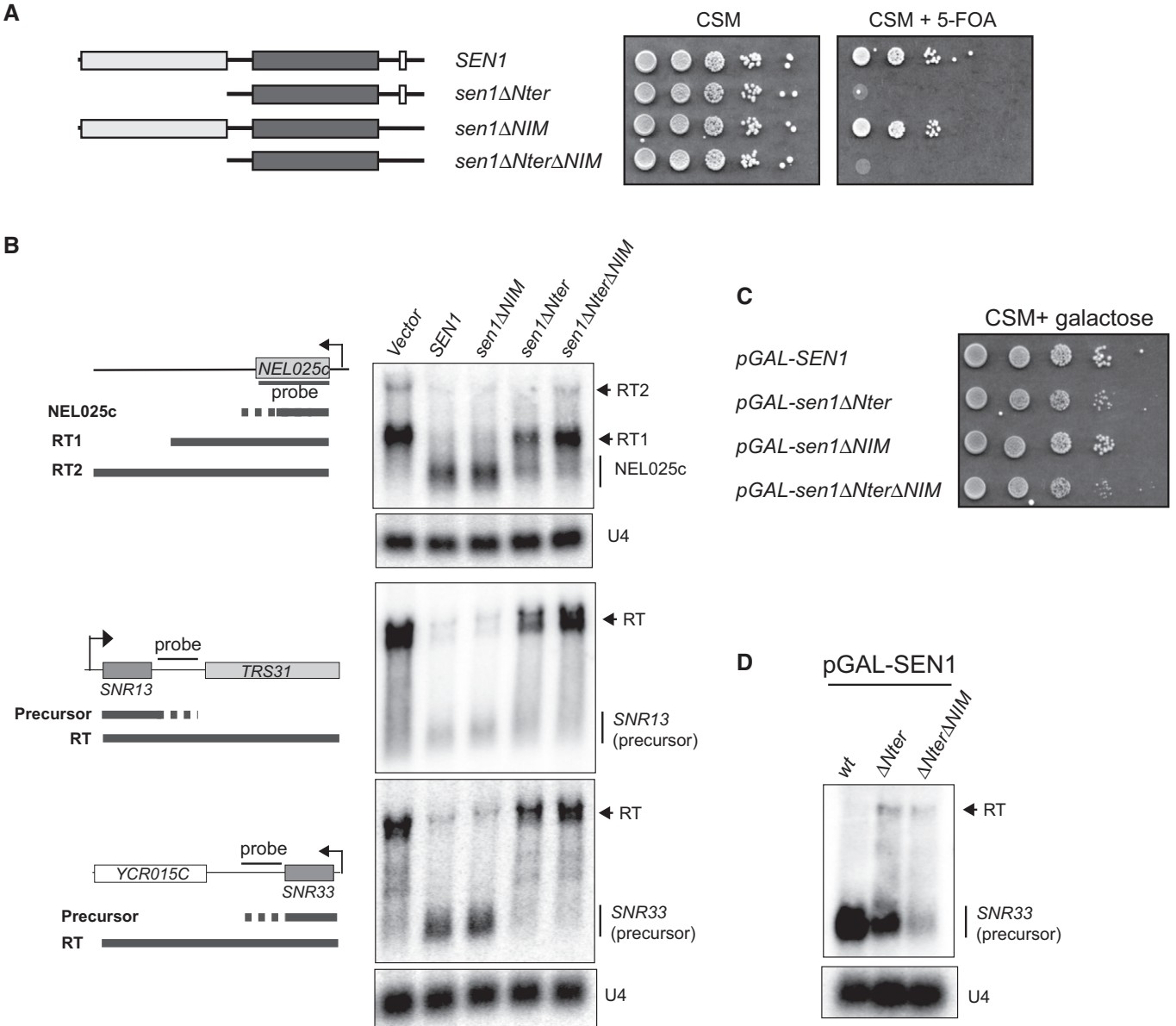

**Figure 3. The N-terminal domain of Sen1 plays an essential role in non-coding transcription termination *in vivo*.**

A Deletion of the N-terminal of Sen1 is lethal. A Δ*sen1* strain (YDL2767) covered by an *URA3*-containing plasmid (pFL38) expressing wt Sen1 was transformed with a *TRP1*-plasmid (pFL39) carrying either the wt or the mutant versions of *SEN1* indicated in the scheme on the left. After overnight growth in non-selective medium, cells were plated on minimal medium (CSM) containing 5-fluorootic acid (5-FOA) to select those that have lost the *URA3* plasmid. The absence of growth in 5-FOA implies that the *SEN1* version expressed in the *TRP1*-plasmid does not support viability.

B Deletion of the N-terminal domain of Sen1 provokes dramatic defects in transcription termination *in vivo*. Northern blot analyses of three well-characterized NNS targets, the CUT NEL025c and the snoRNAs *SNR13* and *SNR33*, in a Sen1-AID (auxin-induced degron) strain carrying an empty vector, or a plasmid expressing either the wt or the indicated versions of *SEN1*. Sen1-AID was depleted for 2 h by the addition of 500 μM indole-3-acetic acid (IAA) to monitor the capacity of the plasmid-borne versions of *SEN1* to induce transcription termination. The expected RNA species resulting from inefficient termination (RT for read-through species) are indicated.

C Overexpression of *sen1ΔNter* restores viability. Growth test of strains harbouring the indicated version of *SEN1* at the endogenous locus under the control of the *GAL1* promoter (pGAL) in the presence of galactose.

D Overexpression of *SEN1* partially suppresses the termination defects associated with deletion of the N-terminal domain. Northern blot analysis of a typical NNS target in strains expressing the indicated *SEN1* versions from pGAL in the presence of galactose.

Data information: For (B) and (D), assays were performed in a Δ*rrp6* background to detect the primary products of NNS-dependent termination and the U4 RNA is used as a loading control. All results shown were reproduced in at least one additional independent experiment. Probes used for RNA detection are described in Appendix Table S6.

2019). However, northern blot analyses of several typical NNS targets revealed only minor transcription termination defects in the *sen1ΔCter* mutant compared to the wt (Fig EV3F).

Taken together, our results indicate that the interaction between Sen1 and Nrd1-Nab3 is not a strict requirement for NNS-dependent termination.

## The integrity of the N-terminal domain of Sen1 is essential for growth and for transcription termination

Our observation that the interaction between Sen1 and Nrd1 and Nab3 is not essential for transcription termination is surprising in the light of the model that posits the Nrd1 and Nab3-dependent recruitment of Sen1. Because Sen1 is a low-abundance protein that binds RNA without any sequence specificity, it seems unlikely that it could be adequately recruited to its targets solely by virtue of its RNA-binding activity. This suggests that additional protein–protein interactions might ensure timely recruitment of Sen1 to the elongation complex. A good candidate to mediate such interactions is the large Sen1 N-terminal domain (aa 1–975), which has been proposed to interact with RNAPII (Ursic *et al*, 2004; Chinchilla *et al*, 2012). In order to explore the possible role of the N-terminal domain (Nter) in Sen1 recruitment, we constructed mutants that carry the deletion of this domain (*sen1ΔNter*) alone or in combination with the NIM deletion (*sen1ΔNterΔNIM*). To our surprise, contrary to reports from two other groups that only observed slow growth (Ursic *et al*, 2004; Chen *et al*, 2014), we found that deletion of Sen1 N-terminal domain was lethal, both when the mutant gene was expressed from a centromeric plasmid (Fig 3A) or from the endogenous locus (Appendix Fig S1). In order to assess the role of the N-terminal domain of Sen1 in NNS-dependent transcription termination, we constructed a Sen1 auxin-inducible degron (AID, Nishimura *et al*, 2009) strain that allowed us to rapidly target Sen1 for degradation by the proteasome upon addition of auxin (Appendix Fig S1). We supplemented the Sen1-AID strain with an empty vector, as a control, or a centromeric plasmid expressing either the wt, ΔNIM, ΔNter or the ΔNterΔNIM version of *SEN1* and we analysed by northern blot the expression of several typical NNS targets upon depletion of the chromosomally encoded copy of *SEN1* (Fig 3B). Strikingly, transcription termination was dramatically impaired in the strain expressing *sen1ΔNter*. In addition, these termination defects were exacerbated in the *sen1ΔNterΔNIM* double mutant. These phenotypes were not due to lower levels of the mutant proteins deleted in the N-terminal domain, since these proteins were expressed at similar levels compared to full-length Sen1 (Appendix Fig S1). Taken together, these results indicate that the N-terminal domain of Sen1 plays a critical role in transcription termination.

The requirement of Sen1 Nter for transcription termination could be explained either by a role of this domain in activating Sen1 catalytic activity or by a function in mediating its interaction with the elongation complex. Because we have previously shown that deletion of the Nter does not affect any of Sen1 measurable catalytic activities (Han *et al*, 2017), we favour the second possibility. In somewhat agreement with this hypothesis, we found that overexpression of *sen1ΔNter* from the *GAL1* promoter (pGAL) restored cell growth (Fig 3C) and partially suppressed the termination defects associated with deletion of Sen1 Nter (Fig 3D).

We set out to explore at the genome-wide level the function of the Nter and the NIM in termination at the different NNS targets. To this end, we performed UV crosslinking and analysis of cDNA (CRAC, Granneman *et al*, 2009) to generate high-resolution maps of transcribing RNAPII on two independent biological replicates of a Sen1-AID strain expressing either the wt, the ΔNIM, the ΔNter or the ΔNterΔNIM version of Sen1 on a plasmid, upon deletion of the chromosomally encoded version of Sen1. In order to have near single nucleotide resolution, we mapped only the 3′ end of reads and we smoothed the signal with a Gaussian kernel (Appendix Fig S2, see methods for details). As expected, none of the Sen1 mutations affected transcription termination at the 3′ end of protein-coding genes (Fig 4A and B, and Appendix Fig S2). However, we detected defects in premature termination at genes that were previously reported to be repressed by attenuation by the NNS-dependent pathway (Fig 4C and D). Whereas the NIM deletion provoked some mild delay in premature termination at the *NRD1* gene (Fig 4C), deletion of the Nter induced strong termination defects at both *NRD1* and *IMD2* (Fig 4C and D). Metagene analyses of CUTs did not unveil any major change in the distribution of transcribing RNAPII in the ΔNIM mutant (Fig 4E–H and Appendix Fig S2), consistent with the results of our former RNAseq experiments (Fig EV3). A more detailed analysis of individual CUTs revealed significant (*P*-value < 0.05, see Table EV1) termination defects at only a minority of CUTs (≈ 7%, see Figs 4I and EV4). In contrast, we observed a substantial increase in the RNAPII occupancy downstream of the annotated termination site of CUTs upon deletion of the Nter, suggesting that deletion of the Nter provokes global transcription termination defects at this class of ncRNAs (see Figs 4E–F and EV4A–D and Appendix Fig S2). The increase in RNAPII signal was even more pronounced in the double ΔNterΔNIM mutant, reflecting exacerbated termination defects at a subset of CUTs (Fig EV4). Altogether, we detected significant termination defects at ≈ 61 and ≈ 68% of CUTs in the ΔNter and the ΔNterΔNIM mutants, respectively (Fig 4I). We observed a very similar trend at snoRNAs (Figs 4J–M and EV4E–H and Appendix Fig S2), with ≈ 8, ≈ 88 and ≈ 90% of snoRNAs being defective in transcription termination in the ΔNIM, the ΔNter and the ΔNterΔNIM mutants, respectively (Fig 4N and Table EV2).

Taken together, these results indicate that the Nter of Sen1 is a critical requirement for transcription termination at most NNS targets. In addition, the protein interactions mediated by the NIM are important for the efficiency of termination at a subset of non-coding genes.

## The N-terminal domain of Sen1 binds preferentially the S5P-CTD of RNAPII

As evoked above, Sen1 Nter has been proposed to interact with RNAPII, more specifically with the S2P form of the CTD (Chinchilla *et al*, 2012), which could be the main way to recruit Sen1 to elongation complexes. We set out to further test this possibility by performing *in vivo* coimmunoprecipitation experiments with *sen1ΔNter* either expressed from its own promoter in a centromeric plasmid using the Sen1-AID system described above or overexpressed from pGAL (Fig 5A and B). Strikingly, we did not detect any significant decrease in the capacity of Sen1 to interact with total RNAPII (shown by Rpb1 and/or Rpb3 subunits) upon

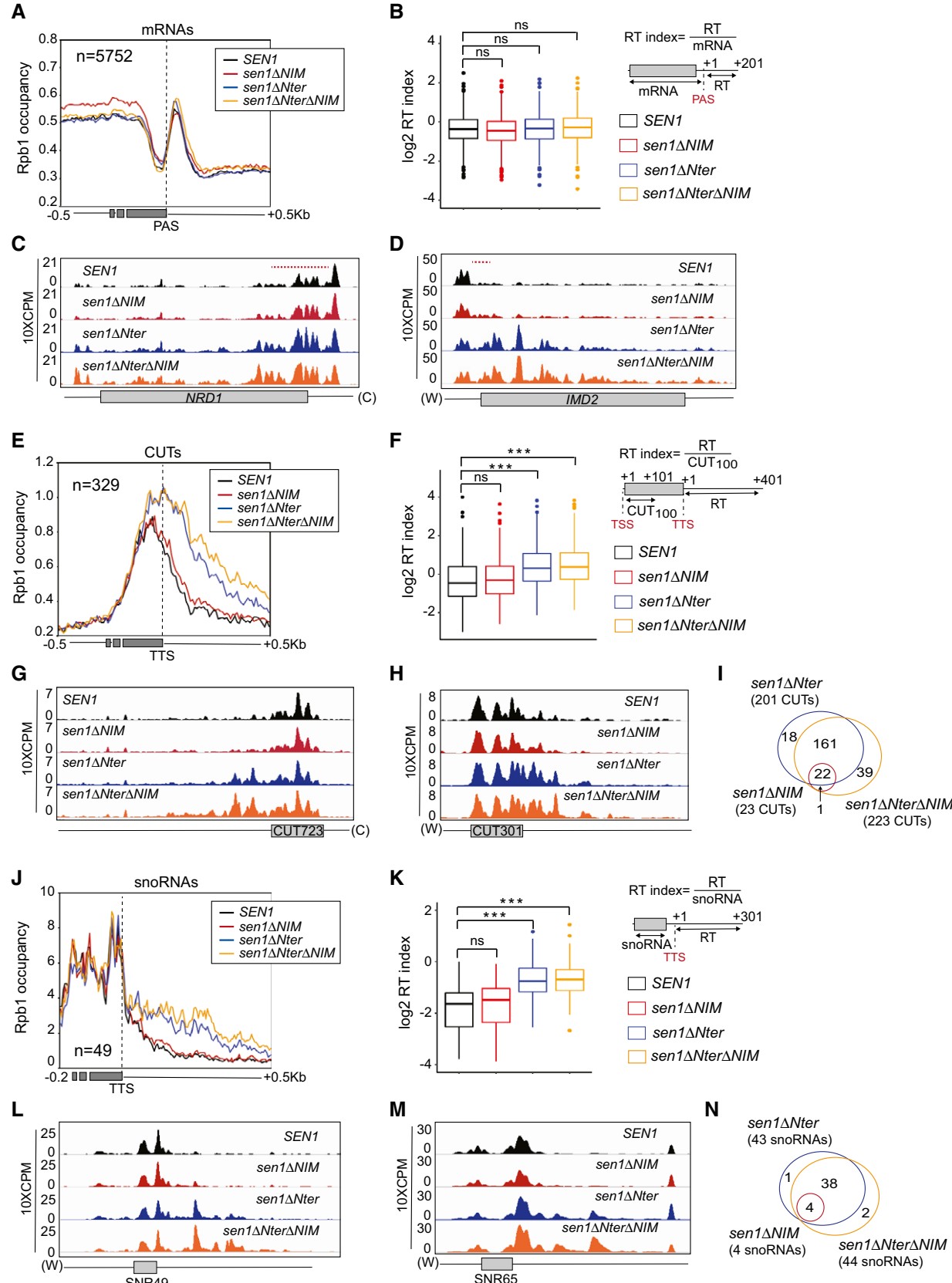

**Figure 4.**

**Figure 4.  High-resolution genome-wide maps of transcribing RNAPII in strains expressing different Sen1 variants.**

RNAPII CRAC was performed in the presence of the indicated versions of *SEN1* expressed from its own promoter in a centromeric plasmid, upon depletion of the chromosomally encoded version of Sen1 in a Sen1-AID strain.

A      Metagene analysis of the RNAPII distribution relative to the annotated polyadenylation site (PAS) of mRNAs. Values on the *y*-axis correspond to the median coverage. Additional analyses performed using the mean and using quartiles are included in Appendix Fig S2.

B      Box-plot representation of the read-through (RT) index as a readout of the termination efficiency at each mRNA. The RT index was calculated as the average signal over a window of 200 nt downstream of the PAS, divided by the average signal from the TSS to the PAS of each mRNA. Boxes include all the values between the 25[th] and the 75[th] percentiles and the horizontal line indicates the median. The upper error bars indicate the distance between the largest value smaller than or equal to the upper limit of the box + 1.5*IQR (where IQR is the interquartile distance or distance between the first and the third quartile). The lower error bars indicate the distance between the lowest value greater than or equal to the lower limit of the box − 1.5 *IQR. For each mRNA ($n$ = 5,752), the values correspond to the average of two independent biological replicates. Data populations were compared using a *t*-test (bilateral distributions, unpaired data) and *P*-values ≥ 0.05 are indicated as ns (not significant).

C, D   IGV screenshots of two well-characterized protein-coding genes displaying substantial defects in premature termination in some mutants. The values correspond to the counts per million of counts (CPM) multiplied by 10. A red dashed line indicates the region were premature termination occurs according to Roy *et al* (2016).

E      Metagene analyses of transcribing RNAPII at CUTs performed as for mRNAs but using the annotated transcription termination site (TTS) as the reference point.

F      Analysis of termination defects in the different mutants at CUTs. Box-plots performed as in (B). The RT index is calculated as the average signal over a window of 400 nt downstream of the TTS, divided by the average signal over a window of 100 nt downstream from the TSS (CUT$_{100}$). For each CUT ($n$ = 329), the values correspond to the average of two independent biological replicates. *P*-values < 0.001 are indicated by three asterisks.

G, H   IGV screenshots of examples of CUTs displaying substantial termination defects in the mutants.

I      Venn diagrams illustrating the overlap between the CUTs that display statistically significant termination defects (*P*-value ≤ 0.05, see Methods for details) in the different mutants. The size of the circles is not proportional to the number of CUTs.

J–N    Analysis of RNAPII occupancy around snoRNAs performed as for CUTs. Box-plots performed as in (B). The RT index is calculated as the average signal over a window of 300 nt downstream of the TTS, divided by the average signal over the mature snoRNA. For each snoRNA ($n$ = 49), the values correspond to the average of two independent biological replicates. *P*-values < 0.001 are indicated by three asterisks.

deletion of the Nter, regardless the expression levels of the mutant protein. In both kind of experiments, protein extracts were treated with RNase to detect only non-RNA-mediated protein interactions. This result strongly suggests that regions other than the Nter can mediate the interaction between Sen1 and RNAPII. We considered the possibility that Sen1 Nter would be important for the recognition of a specific phosphorylated form of RNAPII. Indeed, analyses of Sen1 coimmunoprecipitates with antibodies against different CTD phospho-marks revealed a substantial decrease in the association of S5P-CTD RNAPII with Sen1ΔNter, whereas the levels of bound S2P (Fig 5B), S7P, T4P or unphosphorylated CTD (Appendix Fig S3A) did not change significantly. This result is in contrast with previous two-hybrid assays suggesting that the Sen1 Nter would interact with the S2P form of the CTD (Chinchilla *et al*, 2012) (see Discussion for possible explanations). In order to further substantiate the notion that Sen1 Nter is critical for the recognition of S5P-CTD, we tested whether replacement of this protein region by another domain that can also interact with S5P-CTD could suppress the lethality and termination defects associated with deletion of Sen1 Nter. To this end, we constructed a chimera in which the Nrd1 CID (aa 1–150), which recognizes preferentially the S5P form of RNAPII CTD (Kubicek *et al*, 2012), was fused to Sen1 aa 976–2,231 (i.e. equivalent to Sen1ΔNter). As a control, we also constructed a Sen1 variant carrying the CID domain of Pcf11 (aa 1–137), which recognizes mainly the S2P-CTD (Lunde *et al*, 2010). Interestingly, we found that the strain producing the chimeric Nrd1 CID-Sen1ΔNter protein become viable (Fig 5C) and northern blot analysis of typical NNS targets showed that the efficiency of transcription termination was also restored to a large extent in this strain (Fig 5D). In contrast, the presence of Pcf11 CID at the place of Sen1 Nter did not suppress the growth and transcription termination defects to the same extent (Fig 5C and D), despite both chimeric proteins being expressed at roughly the same levels (Appendix Fig S3B). Taken together, these results strongly support the idea that the essential role of Sen1 Nter is to

mediate the interaction of Sen1 with the CTD of RNAPII and that this domain has a preference for the S5P-CTD.

## The N-terminal and the C-terminal domains of Sen1 can mediate intramolecular interactions

Because the NIM mimics the phosphorylated CTD and our results strongly suggest that the Nter of Sen1 can recognize the S5P-CTD, we considered the possibility that the Nter of Sen1 could interact with the NIM. In order to test this hypothesis, we conducted *in vitro* pull-down experiments with recombinant Sen1 C-terminal domain (aa 1,930–2,231) and the Nter expressed in yeast (Fig 5E). Indeed, we observed substantial and reproducible interaction between both domains of Sen1, indicating that these protein regions have the potential to mediate intramolecular interactions. Strikingly, the Nter could bind equally well the wt and the ΔNIM version of Sen1 Cter, indicating that the Nter recognizes regions other than the NIM in the C-terminal domain.

## High-resolution transcriptome-wide interaction maps of Sen1 variants

Because the S5P-CTD is the most abundant form of RNAPII at short ncRNAs targeted by the NNS complex, the results above suggest that the interaction of Sen1 with the S5P-CTD could be important for the early recruitment of Sen1. Alternatively, this interaction might play a role in a subsequent step, for instance during dissociation of the elongation complex. In order to distinguish between these possibilities and investigate the contribution of the protein interactions mediated by the Nter and the NIM in Sen1 recruitment, we compare the interaction of different Sen1 versions (wt, ΔNIM or ΔNter) with RNA transcripts genome-wide by CRAC (see Methods for details). To monitor for possible contaminations with unbound RNAs, we included as a control a wt strain where Sen1 lacks the tag that allows its purification. The three variants of Sen1 were purified

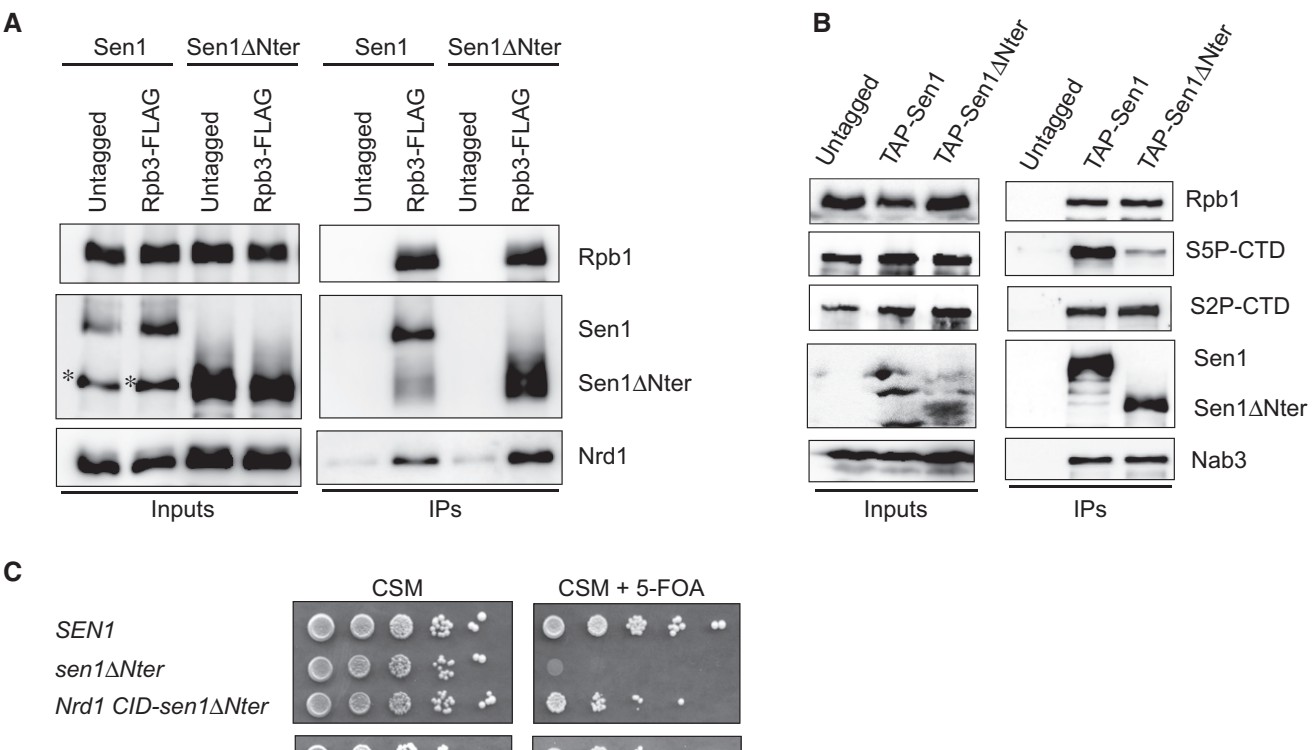

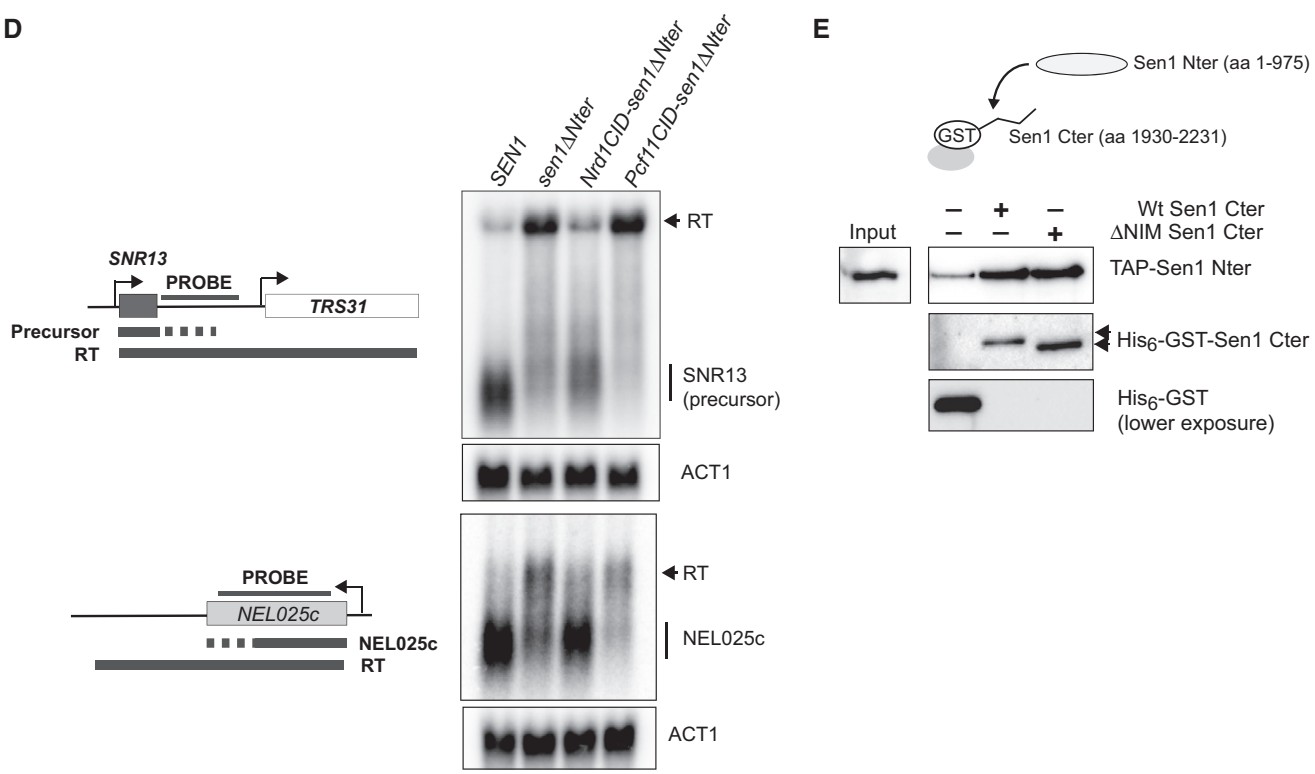

**Figure 5.**

◄

**Figure 5.** **The N-terminal domain of Sen1 can recognize the S5-phosphorylated form of RNAPII CTD and Sen1 C-terminal domain.**

A Deletion of Sen1 N-terminal domain does not prevent the interaction of Sen1 with RNAPII. CoIP experiments using Rbp3-FLAG as the bait. Assays were performed in a Sen1-AID strain harbouring a plasmid expressing either *SEN1* or *sen1ΔNter* upon depletion of Sen1-AID in the presence of IAA for 2 h. An asterisk denotes a major proteolytic Sen1 fragment detected in the extracts of roughly the size of *sen1ΔNter*. Nrd1 is detected as a positive control. Representative gel of one out of two independent experiments. Protein extracts were treated with RNaseA before immunoprecipitation.

C Replacing the Nter of Sen1 by the CID of Nrd1 restores viability. Growth test performed in the same conditions as in Fig 3A but in the presence of a *TRP1*-plasmid carrying the *SEN1* versions indicated in the scheme on the left. The growth of the strain expressing the Nrd1 CID-sen1ΔNter chimera in 5-FOA implies that this gene can support viability.

E Sen1 Nter interacts with the C-terminal domain (Cter) of Sen1 both in the presence and in the absence of the NIM *in vitro*. Pull-down experiments using either a wt or a ΔNIM version of recombinant Sen1 Cter immobilized on glutathione sepharose beads and a TAP-tagged version of Sen1 Nter expressed in yeast. Representative gel of one out of three independent experiments.

at very similar levels (Appendix Fig S4), excluding artefactual results due to different protein yields. The position of the crosslink between the RNA and the protein of interest on the RNA can be revealed by the presence of a deletion or a mutation that is generated during the reverse transcription process. To increase the resolution of our datasets, we mapped only the reads deletions and we subsequently smoothed the signal as for RNAPII CRAC datasets (Appendix Fig S2A, see Methods for details). The signal obtained with the untagged control was very low and spread, unlike the signal of the tagged samples that generated characteristic patterns, arguing for the high quality and specificity of our Sen1 datasets (Fig EV5A and B).

Contrary to the previously reported Sen1 PAR-CLIP data, which did not detect the association of Sen1 with a large fraction of well-characterized NNS targets (Creamer *et al*, 2011), we obtained clear and reproducible signals at all classes of RNAPII transcripts. Comparison of the distribution of Sen1 with that of RNAPII (Rpb1), revealed a remarkably different pattern for both proteins (Fig 6A–F). At mRNAs, RNAPII binding is higher near the transcription start site (TSS), decreases right before the polyadenylation site (PAS) and generates a small peak just after the PAS. This trend was already observed in a former study (Candelli *et al*, 2018). In contrast, Sen1 occupancy is relatively low and homogeneous along most of the mRNA but increases near the 3′ end to form a sharp peak just before the PAS (Fig 6A, B, D and E). Heatmap analyses indicate that this characteristic pattern of Sen1 binding is not restricted to a subset of mRNAs but is rather general (Fig EV5C). The only clear exceptions to this rule would be genes that are regulated by NNS-dependent premature termination, where we observed a strong Sen1 signal also at the 5′ end of the mRNA (Fig 6H and I). At CUTs, Sen1 initial binding seems to be displaced around 50 nt downstream of the RNAPII signal, possibly indicating that the presence of other proteins (e.g. the CAP-binding complex, see Tuck & Tollervey, 2013) tightly bound to the 5′ end of CUTs precludes the interaction of Sen1 with this portion of the RNA (Fig 6C and F). In addition, while the RNAPII occupancy is maximal near the TSS, Sen1 peaks closer to the annotated transcription termination site (TTS, see Figs 6C and EV5D). Accurate comparison between the binding pattern of Sen1 and RNAPII at snoRNAs was complicated by the fact that mature snoRNAs, as other highly abundant RNAs like rRNAs

and tRNAs, are common contaminants in this kind of experiments and tend to pollute more the samples from low-abundance proteins like Sen1 than from RNAPII. Therefore, the Sen1 signal we detect along the mature snoRNAs is not fully reliable. Nevertheless, we detected clear Sen1 binding downstream of the 3′ end of mature snoRNAs (see below).

In order to assess the role of the NIM and the Nter in Sen1 recruitment, we compared the occupancy of the relevant classes of RNAPII transcripts by the different Sen1 variants (Fig 6G–P and Appendix Fig S5A–C). We normalized the Sen1 signal by the RNAPII signal in the corresponding background to correct for the variations in the levels of nascent RNAs due to differences in the termination efficiency by each Sen1 version. Neither deletion of the NIM nor deletion of the Nter affected significantly the binding of Sen1 to mRNAs (Fig 6G and Appendix Fig S5A and D), with the exception of those corresponding to genes subjected to premature termination (Fig 6H and I), which followed the same trend as other typical NNS targets (see below). This indicates that in general these Sen1 regions are not involved in the recruitment of Sen1 to mRNAs. In contrast, the ΔNIM version of Sen1 exhibited a moderate but significant decrease in the association with both CUTs and snoRNAs relative to wt Sen1 (Fig 6J–P). It is important to note that termination by the NNS pathway occurs at multiple sites although only one termination site, typically an early one, is annotated (Roy *et al*, 2016). Therefore, the actual termination region of CUTs and snoRNAs can extend up to few hundred nucleotides downstream of the annotated TTS. Regarding CUTs, reduced Sen1ΔNIM occupancy was observed both at the body and the termination region (Fig 6J–L and Appendix Fig S5B and E). In the case of snoRNAs, we detected decreased RNA binding at the termination region (snoRNA-ter, see Fig 6M–O). These results indicate that the interaction of Sen1 with Nrd1 via the NIM is important for fully efficient recruitment of Sen1 to target ncRNAs.

Surprisingly, despite the strong termination defects provoked by deletion of the Nter (Figs 4 and 5), we did not observe any decrease in the association of Sen1ΔNter with NNS-dependent transcripts compared to the wt (Fig 6J–P and Appendix Fig S5B and E). In fact, we found that Sen1ΔNter progressively accumulated downstream of the annotated TTS, in the case of CUTs, and downstream of the typical termination region in the case of snoRNAs. The fact that the

Sen1ΔNter protein accumulates along the terminator read-through region suggests that this accumulation is a consequence rather than a cause of the transcription termination defects observed in the mutant. These results imply that the interaction of Sen1 with the CTD of RNAPII should function downstream of the initial recruitment of Sen1 to ncRNAs.

## Decreasing the RNAPII elongation rate bypasses the requirement of the Sen1 N-terminal domain for termination

The above evidence suggests that the interaction between Sen1 Nter and the RNAPII CTD would play an important role during dissociation of the transcription elongation complex. This is

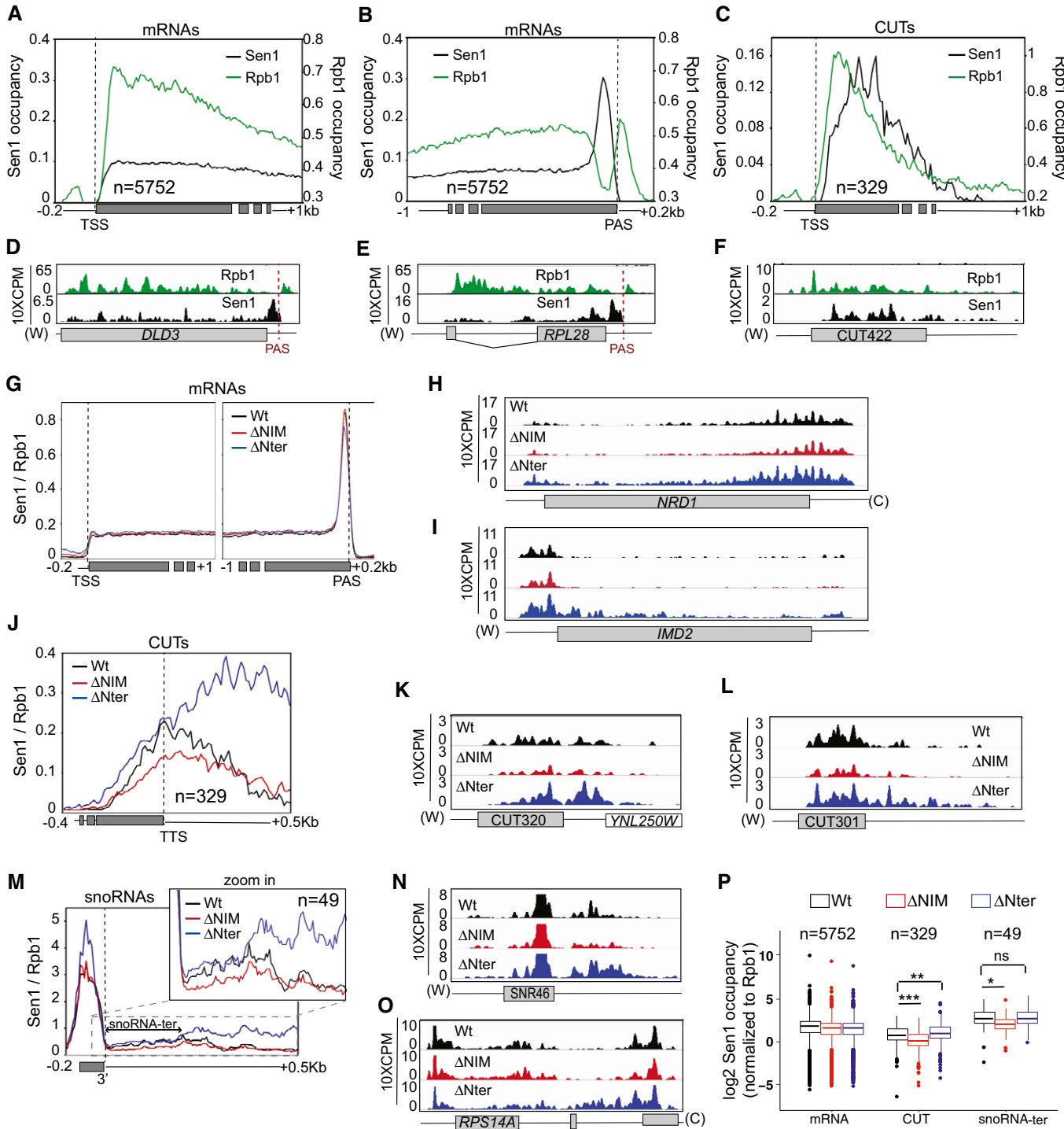

**Figure 6.**

**Figure 6. High-resolution transcriptome-wide maps of binding sites for different Sen1 variants.**

A–C Comparison of the distribution of Sen1 and RNAPII (Rpb1) at mRNAs and CUTs. Values on the *y*-axis correspond to the median coverage. Note that the scale of Sen1 and Rpb1 datasets has been modified to be able to visualize both in the same plot.

D–F IGV screenshots of two mRNAs and one CUT exemplifying the trend observed in the metagene plots.

G Metagene analysis of the distribution of the different Sen1 variants at mRNAs using either the TSS or the PAS as reference point. Values correspond to the median of Sen1 data normalized by the RNAPII signal obtained in the corresponding background. Additional analyses performed using the mean and using quartiles are included in Appendix Fig S5.

H, I Screenshots of two well-characterized protein-coding genes that are regulated by NNS-dependent premature termination. Values correspond to unnormalized Sen1 occupancy.

J Profile of Sen1 binding to CUTs calculated as for mRNAs.

K, L Examples of CUTs exhibiting differences in Sen1 occupancy upon deletion of the NIM or the Nter. Plots correspond to unnormalized data.

M–O Analysis of the distribution of the different Sen1 versions around snoRNAs performed as for CUTs.

P Comparison of the total levels of the different Sen1 variants at mRNAs, CUTs and the termination region of snoRNAs (i.e. a window of 200 nt downstream of the 3′ end of the mature snoRNA, snoRNA-ter). Boxes in box-plots include all the values between the 25th and the 75th percentiles, and the horizontal line indicates the median. The upper error bars indicate the distance between the largest value smaller than or equal to the upper limit of the box + 1.5*IQR. The lower error bars indicate the distance between the lowest value greater than or equal to the lower limit of the box − 1.5*IQR. Datasets for the different Sen1 versions were compared using a *t*-test (bilateral distributions, unpaired data). *P*-values ≥ 0.05 are indicated as ns (not significant), one asterisk is used when 0.05 > *P*-value ≥ 0.01, two asterisks when 0.01 > *P*-value ≥ 0.001 and three asterisks for *P*-value < 0.001.

striking in the light of our former data showing that *in vitro* neither deletion of the CTD nor deletion of the Nter affected significantly the efficiency of elongation complex dismantling (Porrua & Libri, 2013; Han *et al*, 2017). We noticed, however, that in our *in vitro* system Sen1-mediated termination strictly required RNAPII to be stalled by either the presence of a roadblock protein or by the removal of one of the four nucleotides. This process can be partially mimicked *in vivo* by the addition of 6-azauracil (6-AU), which decreases the intracellular concentration of GTP and UTP, therefore strongly reducing RNAPII elongation rate (Exinger & Lacroute, 1992). We set out to test whether the presence of 6-AU in the media could improve transcription termination by Sen1ΔNter (Fig 7A). Northern blot analysis of the NEL025c ncRNA showed that the addition of 6-AU led to a slight reduction in the size of the RNA species detected in the wt strain, indicating that transcription termination occurred earlier. This is consistent with previous evidence of a kinetic competition between transcription elongation and NNS-dependent termination (Hazelbaker *et al*, 2013). Interestingly, the presence of 6-AU induced a strong suppression of the termination defects provoked by deletion of Sen1 Nter. A similar trend was observed at *SNR33*, although the result is less clear because the 6-AU provoked also a substantial reduction in the RNA signal, as previously observed (Hazelbaker *et al*, 2013). These results suggest that the interaction between Sen1 Nter and the RNAPII CTD provides an advantage to Sen1 in the context of a kinetic competition between termination and transcription elongation.

## Discussion

In budding yeast, the NNS complex emerges as a safeguard for gene expression. On one hand, it is required for transcription termination and maturation of snoRNAs, which in turn have important functions in rRNA modification. On the other hand, via its transcription termination activity coupled to RNA degradation, it prevents massive genomic deregulation that results from uncontrolled pervasive transcription (Schulz *et al*, 2013). Nevertheless, the NNS complex needs to be tightly regulated in order to restrict its activity to the right targets and avoid premature transcription termination at protein-coding genes and/or degradation of mRNAs. The final step of transcription termination (i.e. dissociation of the elongation complex) exclusively depends on the helicase Sen1, which cannot discriminate non-coding from protein-coding RNAs on its own. The fact that Sen1 forms a complex with Nrd1 and Nab3, together with the capacity of Nrd1 and Nab3 to recognize motifs that are enriched in the target ncRNAs, led several authors, including us, to propose that Nrd1 and Nab3 would play a critical role in the recruitment of Sen1, therefore conferring the necessary specificity to Sen1 activity.

In the present study, we characterize molecularly and functionally the interaction between Sen1 and its partners Nrd1 and Nab3 and provide data that challenge the former model. Furthermore, we show that the interaction between Sen1 and the CTD of RNAPII is a critical requirement for the action of Sen1 on ncRNA genes. Our results allow redefining the rules that govern the specific function of the NNS complex in non-coding transcription termination.

**Figure 7. Influence of the transcription elongation rate on the requirement of Sen1 Nter for non-coding transcription termination.**

A Decreasing the elongation rate alleviates the termination defects associated with deletion of Sen1 Nter. Northern blot assays performed in a Sen1-AID, Δ*rrp6* strain carrying a plasmid expressing the indicated versions of *SEN1* upon depletion of the endogenous Sen1 protein as in former experiments. Where indicated, cells were treated with 50 mg/l of 6-azauracil (6AU) for 2 h. Representative gel of one out of two independent biological replicates. The U4 RNA is used as a loading control. Probes used for RNA detection are described in Appendix Table S6.

B Model for the role of the N-terminal domain and the NIM in the control of Sen1 action at ncRNAs. Nrd1 and Sen1 are likely independently recruited to the elongation complex. The recruitment of Nrd1 and Nab3 is mediated by the recognition of specific RNA sequences and enhanced by the interaction of Nrd1 CID with the S5P-CTD via the CID. Sen1 would be recruited either by unspecific binding to the nascent RNA or by possibly transient protein–protein interactions with RNAPII. The interaction of Sen1 NIM with Nrd1 CID also enhances Sen1 recruitment. The interaction of Sen1 Nter with the CTD could facilitate the loading of Sen1 to the nascent RNA in the vicinity of paused RNAPII. Sen1 would translocate along the nascent RNA to induce dissociation of the elongation complex. The release of Sen1 from Nrd1 would be required for the CID to be available to recruit TRAMP for subsequent processing/degradation of the released RNA.

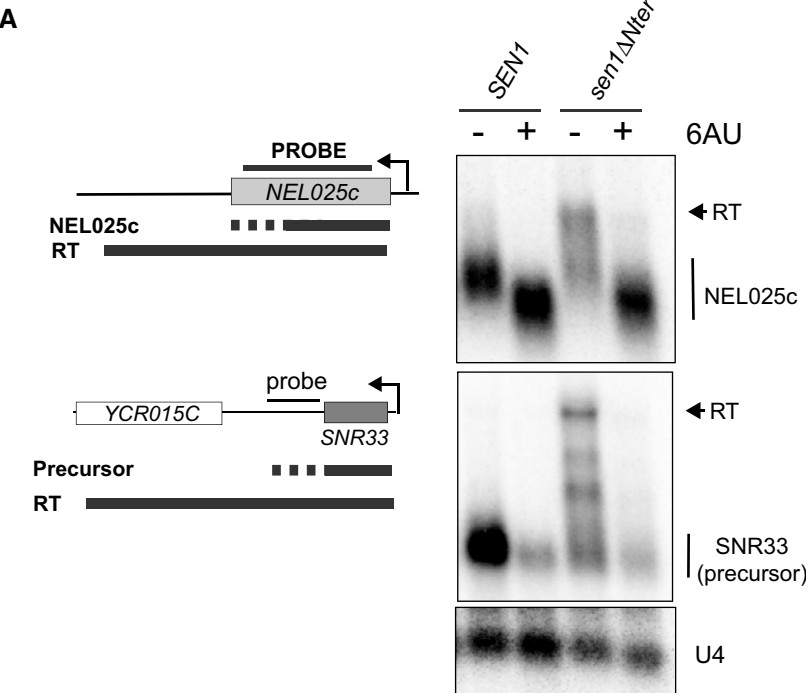

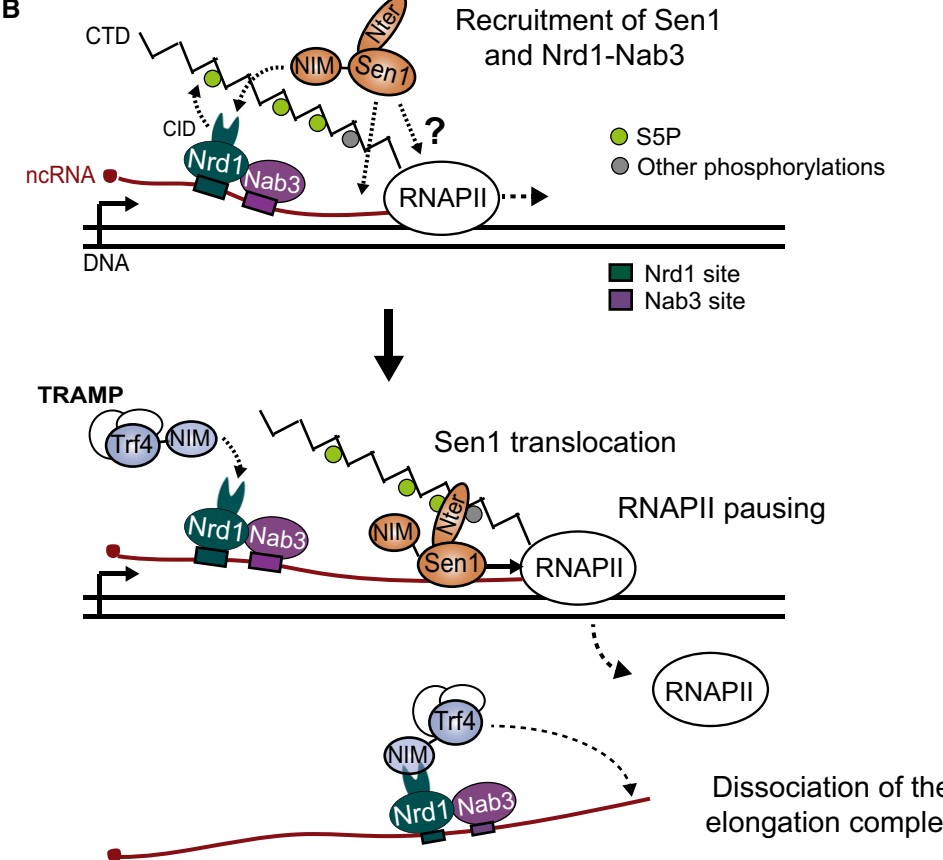

**Figure 7.**

## Molecular mimicry to coordinate transcriptional and post-transcriptional processes

As mentioned above, the CTD of RNAPII is considered a master regulator of the transcription cycle. This is due to the capacity of the CTD to be dynamically modified by kinases and phosphatases resulting in complex phosphorylation patterns that are differentially recognized by a plethora of factors with key roles in transcription-related processes (Harlen & Churchman, 2017a,b). One of these factors is Nrd1, that preferentially binds the S5P-CTD to induce early transcription termination (Vasiljeva *et al*, 2008; Kubicek *et al*, 2012). Surprisingly, in a previous study, we have discovered that the same domain of Nrd1 recognizes a sequence that mimics the S5P-CTD in the non-canonical poly(A)-polymerase Trf4. In a subsequent report, a second CTD mimic was identified in another cofactor of the exosome, Mpp6 (Kim *et al*, 2016). Here, we reveal the presence of a third functional CTD mimic, designated NIM for Nrd1-Interaction Motif, in the essential helicase Sen1. This mechanism of mediating mutually exclusive interactions with multiple partners that should act sequentially in the same RNA molecule seems an efficient manner to temporally coordinate the different steps of the NNS pathway: recruitment of Nrd1 to RNAPII, then recruitment/stabilization of Sen1 to facilitate dissociation of the transcription elongation complex and finally polyadenylation of the released RNAs by Trf4 and subsequent degradation/processing by the exosome (Fig 7B).

Our previous structural analyses have shown that the NIM in Trf4 shares with the S5P-CTD several major structural elements: one or several negatively charged residues at the N-terminal part and several hydrophobic aa that adopt a β-turn at the C-terminal part, flanked by a conserved tyrosine. Strikingly, in the present structure of Nrd1 CID in complex with Sen1 NIM we have not observed the characteristic β-turn, yet several important H-bonds and hydrophobic interactions between the CID and Sen1 NIM are maintained and the affinity of the CID for the NIMs of Trf4 and Sen1 is almost identical. This is due to an alternative conformation of the C-terminal region of Sen1 NIM that can be accommodated in the binding pocket of Nrd1 CID. We suggest that similar extended conformation exists also in the case of the CTD mimic found in Mpp6 (Kim *et al*, 2016) as its DLDK C-terminal motif (Fig 2E) has no propensity to form a β-turn (Singh *et al*, 2015).

This relaxed sequence requirement implies that other protein regions could potentially behave as *bona fide* CTD mimics mediating the interaction with Nrd1 or with other factors with structurally related CIDs. Of course, such protein regions should be present in the appropriate protein context (i.e. intrinsically disorder regions and not covered by other protein–protein interaction regions) and in the appropriate cellular compartment (i.e. in the nucleus, in the case of Nrd1-interacting factors). We anticipate that other proteins that function both in association with the CTD of RNAPII and in a separate complex might employ a similar mechanism to either coordinate transcriptional and post-transcriptional steps of the same pathway, as in the case of Nrd1, or to mediate independent functions of the same protein (e.g. docking of a CTD kinase to a different substrate).

## Several ways to recruit Sen1 to transcribing RNAPII

In order to understand how the different protein–protein interactions characterized in this study may partake in the recruitment of Sen1 to nascent transcripts, we have generated high-resolution transcriptome-wide maps of Sen1 binding sites (Fig 6). We have detected a clear and reproducible interaction of Sen1 with all classes of RNAPII transcripts. After normalization to RNAPII occupancy, which is proportional to transcription levels, we have found that the levels of Sen1 at mRNAs, CUTs and the termination region of snoRNAs are actually within the same range.

Our data show that the interaction of Sen1 with its partners Nrd1 and Nab3 via the NIM specifically and generally contributes to the recruitment of Sen1 to ncRNAs (Fig 6). However, this contribution is rather moderate, as we have observed only a ≤ 2-fold decrease in Sen1 occupancy upon deletion of the NIM. In addition, this reduction in Sen1 levels only has a significant impact on the efficiency of termination at a small fraction of NNS targets, although the effects of the NIM deletion are more severe when Sen1 also lacks its N-terminal domain (Fig 4). Furthermore, full deletion of the C-terminal domain of Sen1 completely abolishes the interaction between Sen1 and its partners but has only a minor impact on transcription termination efficiency (Fig EV3). This suggests that there might be a minimal level of Sen1 that can support efficient termination and that in most instances preventing the interaction with Nrd1 and Nab3 does not decrease Sen1 levels below that threshold. However, more importantly, our results indicate that there should be additional mechanisms to recruit Sen1 to ncRNAs that do not involve the interaction with Nrd1 and Nab3. Because Sen1 can interact with essentially all mRNAs, regardless the presence of RNA-bound Nrd1 and Nab3, it is likely that there is a common and essential mechanism of Sen1 recruitment that operates at both protein-coding and non-coding transcripts. This mechanism might involve the direct interaction of Sen1 with RNAPII. The Sen1 surfaces involved in this possible interaction could be located at the helicase domain because we have observed substantial and reproducible interaction between Sen1 and RNAPII even in the absence of the N-terminal domain of Sen1 (Fig 5). One could argue that the C-terminal domain might mediate this interaction, but the weak phenotype of a Sen1 mutant lacking the whole Cter makes this possibility unlikely. Nevertheless, the observed interaction between Sen1 and RNAPII could be at least partially RNA-dependent. Indeed, although we include in our assays a treatment with RNase, the RNA bridging the two proteins might not be accessible to the RNase if Sen1 is very tightly bound to the nascent transcript and in very close proximity to RNAPII. In that case, it is possible that Sen1 samples all nuclear RNAs using the sequence-unspecific RNA-binding activity of its helicase domain and it induces transcription termination only when the context is appropriate.

In both scenarios (i.e. direct Sen1 recruitment to elongation complexes via the interaction with either RNAPII or with the nascent RNA), the specificity on ncRNAs would be provided at least in part by the concomitant presence and action of its partners Nrd1 and Nab3. Indeed, Nrd1 and Nab3 are essential and their interaction with specific sequences in the target ncRNAs is critical for termination *in vivo* (Steinmetz & Brow, 1998; Conrad *et al*, 2000; Steinmetz *et al*, 2001, 2006; Carroll *et al*, 2007). It has previously been proposed that Nrd1 and Nab3 would induce RNAPII pausing (Schaughency *et al*, 2014). In a former study, we have provided evidence that Sen1 is a poorly processive translocase and that *in vitro* transcription termination by Sen1 strictly requires RNAPII to be paused (Han *et al*, 2017). Therefore, a model in which Nrd1 and

Nab3 would promote Sen1-dependent termination by modifying RNAPII elongation properties would be consistent with present and previous data. Understanding how the interaction of Nrd1 and Nab3 with the nascent RNAs and/or with RNAPII-associated factors would induce polymerase pausing is an interesting subject for future studies.

## The essential role of Sen1 N-terminal domain in non-coding transcription termination

In the present study, we have shown that deletion of the N-terminal domain of Sen1 is lethal (Fig 3). The essential character of the Nter of Sen1 was overlooked in former studies (Ursic *et al*, 2004; Chen *et al*, 2014), most likely because the *sen1ΔNter* gene was not expressed under its own promoter at the endogenous locus, and therefore possibly higher levels of the truncated protein produced in those genetic contexts would be sufficient for viability. Indeed, we have found that overexpression of *sen1ΔNter* suppresses lethality (Fig 3). In addition, the role of this domain in transcription termination has not systematically been analysed before. Here, we present high-resolution genome-wide data showing that the Nter is a global requirement for transcription termination specifically at non-coding genes (Fig 4).

The results of our biochemical and genetic analyses strongly suggest that the essential function of the Nter is to promote the interaction of Sen1 with the RNAPII CTD and preferentially with the S5P form of the CTD (Fig 5). First, deletion of the Nter decreases the association of Sen1 specifically with the S5P-CTD in coimmunoprecipitation experiments. Second, replacing the Nter of Sen1 by a CID domain that preferentially recognizes the S5P-CTD suppresses both the lethality and the termination defects associated with deletion of the Nter, while a CID that binds mainly the S2P-CTD does not rescue these phenotypes. A previous work reported that Sen1 preferentially binds the S2P form of the CTD in two-hybrid assays (Chinchilla *et al*, 2012). It is possible that in that artificial context, because of a different conformation of the CTD separated from the rest of RNAPII and/or a different repertoire of factors bound to the CTD, Sen1 exhibits a more efficient interaction with the S2P-CTD. We cannot exclude that the Nter of Sen1 can actually bind the S2P-CTD, in addition to the S5P-CTD. However, our results indicate that the capacity to interact with the S2P-CTD is not particularly relevant for the function of Sen1 on non-coding genes. In fact, a very recent genome-wide study has shown that a mutation in Sen1 Nter that has been proposed to impair the recognition of the S2P-CTD (R302W) does not have any impact on NNS-dependent transcription termination (Collin *et al*, 2019). We have obtained similar results on isolated targets (data not shown). However, the same study did not detect any clear termination defect in a strain where the S5 of each CTD repeat has been mutated to alanine, which is in seemingly contrast with the idea that the essential function of Sen1 Nter is to mediate the recognition of the S5P-CTD. The results obtained with such mutant might, nevertheless, be difficult to interpret regarding termination because S5 phosphorylation is critical for promoter escape by RNAPII (Jeronimo & Robert, 2014). Indeed, polymerases that are retained at the promoter cannot be subjected to termination because they have not reached the termination region. We have re-analysed the implication of S5P in termination by assessing the levels of read-through transcripts relative to the transcripts ending at the primary site of termination. To this end, we have performed northern blot analyses on a thermosensitive mutant of Kin28, the main S5 kinase, as well as a strain expressing a S5A version of rpb1 (see Appendix Fig S6). In order to detect the primary products of efficient termination, that are unstable, we have performed these experiments in a Δ*rrp6* background. Both inactivation of Kin28 at 37°C and the presence of the S5A mutations in the CTD provoked a dramatic decrease in the levels of the ncRNAs tested as well as of the *ACT1* mRNA, strongly suggesting a general impairment in the production of full-length RNAs. This supports the notion that S5 phosphorylation is required for RNAPII transition to productive elongation and, therefore, precludes a reliable assessment of the role of S5P in Sen1-dependent termination. Nonetheless, we cannot completely exclude the possibility that other phosphorylated forms of the CTD can also support, to some extent, the interaction with Sen1 Nter and partake in transcription termination (Fig 7B). Structural analyses of the Nter should shed light on this matter. We have also found that Sen1 Nter can bind the C-terminal domain of Sen1 *in trans* (Fig 5). We speculate that this intramolecular interaction might modulate the association of Sen1 N-terminal and C-terminal domains with additional factors (e.g. with the RNAPII and Nrd1).

The global and strong termination defects provoked by the deletion of the Nter prompted us to hypothesize that the interaction of Sen1 with the CTD might be necessary for the recruitment of Sen1 to ncRNAs. However, our Sen1 CRAC experiments support rather a role for this interaction downstream of Sen1 initial recruitment. Indeed, upon deletion of the Nter we do not observe a decrease in Sen1 occupancy but rather a progressive accumulation downstream of the early termination sites. This suggests that, when the mutant protein fails to induce termination, it tracks the transcribing RNAPII in an unproductive manner until RNAPIIs are terminated, possibly by other pathways. This behaviour would resemble to some extent that of Sen1 at mRNAs, where Sen1 is present but does not induce termination and accumulates just before the PAS, i.e. the region where RNAPII would likely get committed to termination by the CPF pathway. We do not understand at this point the role of this prominent presence of Sen1 at the 3′ end of mRNAs. It is possible that it is related to a function for Sen1 as fail-safe terminator at protein-coding genes, as previously proposed (Rondón *et al*, 2009). Further experimental work will be required to elucidate this matter.

Regarding the precise role of the interaction of Sen1 with the CTD during termination, we have shown that this interaction is almost irrelevant when the RNAPII elongation rate is artificially reduced (Fig 7A). A previous study has provided evidence for the kinetic competition between elongation and Sen1-mediated transcription termination (Hazelbaker *et al*, 2013). The same concept has been shown later to apply also to the termination pathway for protein-coding genes in mammals (Fong *et al*, 2015). This notion is supported by the fact that termination is delayed (i.e. occurs downstream of natural termination sites) for fast mutant polymerases, whereas slow mutant polymerases undergo termination earlier. A very recent report has further substantiated this kinetic competition by showing that promoter-proximal RNAPII pausing is required for efficient NNS-dependent termination and depends on phosphorylation of tyrosine 1 of the RNAPII CTD (Collin *et al*, 2019). The mechanisms that connect Y1 phosphorylation and pausing as well as whether they involve the action of Nrd1 and Nab3, as discussed above, remain an exciting subject for future studies. A possible

explanation for the requirement of RNAPII pausing by Sen1-mediated termination is provided by our former observation that *in vitro* Sen1 is a poorly processive translocase, implying that it might undergo frequent cycles of dissociation from and re-association with the nascent transcript before it can productively reach the RNAPII (Han *et al*, 2017). Therefore, polymerase pausing might provide a window of opportunity for Sen1 to induce termination and the interaction of Sen1 Nter with the CTD of RNAPII could be critical to position Sen1 within this window. The use of an elongation inhibitor would, thus, simply extend the window of opportunity for termination (Fig 7A and B), rendering the interaction of Sen1 with the CTD less critical.

In conclusion, because of the particular characteristics of NNS targets as transcription units and the intrinsic properties of Sen1 as a transcription termination factor, the recognition of the CTD by Sen1 N-terminal domain unveiled in this study is, together with the presence of Nrd1 and Nab3 binding sites, an important determinant of the specificity of the NNS complex for short non-coding genes.

# Materials and Methods

### Construction of yeast strains and plasmids

Yeast strains used in this paper are listed in Appendix Table S4. Gene deletions, tagging and insertion of the *GAL1* promoter were performed with standard procedures (Longtine *et al*, 1998; Rigaut *et al*, 1999) using plasmids described in Appendix Table S5. Strain DLY2769 expressing untagged *sen1ΔNIM* was constructed by transforming a *Δsen1 strain* harbouring the *URA3*-containing plasmid pFL38-*SEN1* (DLY2767) with the product of cleavage of pFL39-*sen1ΔNIM* (pDL703) with restriction enzymes MluI, BstZ17I and Bsu36I. Cells capable of growing on 5-FOA were then screened by PCR for the presence of the NIM deletion and the absence of plasmids pFL38-*SEN1* and pFL39-*sen1ΔNIM*.

Plasmids expressing different *SEN1* variants were constructed by homologous recombination in yeast. Briefly, a wt yeast strain was transformed with the corresponding linearized vector and a PCR fragment harbouring the region of interest flanked by 40–45 bp sequences allowing recombination with the vector. Clones were screened by PCR, and the positive ones were verified by sequencing.

Plasmids for overexpression of the Nrd1 CID variants R133G and R133D were obtained using QuikChange site-directed mutagenesis kit (Stratagene).

### Coimmunoprecipitation experiments

Yeast extracts were prepared by standard methods. Briefly, cell pellets were resuspended in lysis buffer (10 mM sodium phosphate pH 7, 200 mM sodium acetate, 0.25% NP-40, 2 mM EDTA, 1 mM EGTA, 5% glycerol) containing protease inhibitors, frozen in liquid nitrogen and lysed using a Retsch MM301 Ball Mill. For TAP-tagged proteins, the protein extract was incubated with IgG Fast Flow Sepharose (GE Healthcare) and beads were washed with lysis buffer. Tagged and associated proteins were eluted either by cleaving the protein A moiety of the tag with TEV protease in cleavage buffer (10 mM Tris pH 8, 150 mM NaCl, 0.1% NP-40, 0.5 mM

EDTA and 1 mM DTT) for 2 h at 21°C or by boiling the beads in 2× Laemmli buffer (100 mM Tris pH 6.8, 4% SDS, 15% glycerol, 25 mM EDTA, 100 mM DTT, 0.2% bromophenol blue) for 5 min. For HA-tagged proteins, protein extracts were first incubated with 25 μg of anti-HA antibody (12CA5) for 2 h at 4°C. Then, 30 μl Protein A-coupled beads (Dynabeads, Thermo Fisher, 30 mg/ml) were added to each sample and incubated for 2 h at 4°C. After incubation, the beads were washed with lysis buffer and proteins were eluted by incubating with 2× Laemmli buffer (without DTT) for 10 min at 37°C.

For pull-down experiments, each recombinant version of His$_6$-GST-tagged Sen1 Cter or the His$_6$-GST control was overexpressed by growing BL21 (DE3) CodonPlus (Stratagene) cells harbouring the appropriate plasmid (see Appendix Table S5) on auto-inducing medium (Studier, 2005) at 20°C overnight. Protein extracts were prepared in GST binding buffer (50 mM Tris–HCl, pH 7.5, 500 mM NaCl, 5% glycerol, 0.1% NP-40, 1 mM DTT) by sonication and subsequent centrifugation at 16,000 *g* for 30 min at 4°C. Approximately 5 mg of extract containing the corresponding recombinant protein was incubated with 25 μl of glutathione sepharose (SIGMA) and subsequently mixed with yeast extracts expressing Sen1 TAP-Nter (typically 0.5 mg of extract per binding reaction). Beads were washed with lysis buffer (10 mM sodium phosphate pH 7, 200 mM sodium acetate, 0.25% NP-40, 2 mM EDTA, 1 mM EGTA, 5% glycerol) and the proteins were eluted by incubation for 15 min in 80 μl GST elution buffer containing 50 mM Tris–HCl pH 8, 20 mM reduced glutathione, 150 mM NaCl, 0.1% NP-40 and 10% glycerol. Protein extracts were treated with 50 μg/ml of RNase A for 20 min at 20°C prior to incubation with beads.

### Fluorescence anisotropy analyses

Nrd1 CID and its mutants were produced and purified as described previously (Kubicek *et al*, 2012). The equilibrium binding of the different versions of Nrd1 CID to Trf4 NIM and Sen1 NIM was analysed by FA. The NIM peptides were N-terminally labelled with the 5,6-carboxyfluorescein (FAM). The measurements were conducted on a FluoroLog-3 spectrofluorometer (Horiba Jobin-Yvon Edison, NJ). The instrument was equipped with a thermostated cell holder with a Neslab RTE7 water bath (Thermo Scientific). Samples were excited with vertically polarized light at 467 nm, and both vertical and horizontal emissions were recorded at 516 nm. All measurements were conducted at 10°C in 50 mM Na$_2$HPO$_4$ 100 mM NaCl pH = 8. Each data point is an average of three measurements. The experimental binding isotherms were analysed by DynaFit using 1:1 model with non-specific binding (Kuzmic, 2009).

### NMR analyses

All NMR spectra were recorded on Bruker AVANCE III HD 950, 850, 700 and 600 MHz spectrometers equipped with cryoprobes at a sample temperature of 20°C using 1 mM uniformly $^{15}$N,$^{13}$C-labelled Nrd1-CID in 50 mM Na$_2$HPO$_4$ 100 mM NaCl pH = 8 (20°C; 90% H$_2$O/10% D$_2$O). The initial nuclei assignment was transferred from BMRB entry 19954 and confirmed by HNCA, HNCACB, HCCCONH, HBHACONH and 4D-HCCH TOCSY spectra. The spectra were processed using TOPSPIN 3.2 (Bruker Biospin), and the protein resonances were assigned manually using Sparky software

(Goddard T.G. and Kellner D.G., University of California, San Francisco). 4D version of HCCH TOCSY (Kay *et al*, 1993) was measured with a non-uniform sampling; acquired data were processed and analysed analogously as described previously (Nováček *et al*, 2011, 2012). All distance constraints were derived from the three-dimensional 15N- and 13C-edited NOESYs collected on a 950 MHz spectrometer. Additionally, intermolecular distance constraints were obtained from the three-dimensional F1-13C/15N-filtered NOESY-[13C,1H]-HSQC experiment (Zwahlen *et al*, 1997; Peterson *et al*, 2004) with a mixing time of 150 ms on a 950 MHz spectrometer. The NOEs were semi-quantitatively classified based on their intensities in the 3D NOESY spectra. The initial structure determinations of the Nrd1 CID–Sen1 NIM complex were performed with the automated NOE assignment module implemented in the CYANA 3.97 program (Güntert & Buchner, 2015). Then, the CYANA-generated restraints along with manually assigned Nrd1 CID–Sen1 NIM intermolecular restraints were used for further refinement of the preliminary structures with AMBER16 software (Case, 2002). These calculations employed a modified version (AMBER ff14SB) of the force field (Maier *et al*, 2015), using a protocol described previously (Stefl *et al*, 2010; Hobor *et al*, 2011). The 20 lowest-energy conformers were selected (out of 50 calculated) to form the final ensemble of structures.

### Northern blot assays

Unless otherwise indicated, cells used for northern blot assays were grown on YPD medium at 30°C to $OD_{600}$ 0.3–0.6 and harvested by centrifugation. RNAs were prepared using standard methods. Samples were separated by electrophoresis on 1.2% agarose gels and then transferred to nitrocellulose membranes and UV-crosslinked. Radiolabelled probes were prepared by random priming of PCR products covering the regions of interest with Megaprime kit (GE Healthcare) in the presence of $\alpha$-$^{32}$P dCTP (3,000 Ci/mmol). Oligonucleotides used to generate the PCR probes are listed in Appendix Table S6. Hybridizations were performed using a commercial buffer (Ultrahyb, Ambion) and after washes, membranes were analysed by phosphorimaging.

### RNAseq library preparation and deep sequencing

RNA samples were treated with RiboZero to deplete rRNA, and RNA libraries were prepared by the IMAGIF sequencing platform using a NextSeq 500/550 High Output Kit v2. Samples were sequenced on a NextSeq 500 sequencer.

### UV crosslinking and analysis of cDNA (CRAC)

For RNAPII, we employed the CRAC procedure reported in Granneman *et al* (2009) with several modifications described in Candelli *et al* (2018). We employed a Sen1-AID strain expressing an HTP-tagged version of the Rpb1 subunit of RNAPII (see Appendix Table S4) at the endogenous locus and different versions of Sen1 on a centromeric plasmid (see Appendix Table S5). Briefly, 2 l of yeast cells were grown at 30°C to $OD_{600} = 0.3$ in CSM-TRP medium before addition of IAA to a final concentration of 500 μM and further incubation for 2 h. Cells were crosslinked for 50 s using a W5 UV crosslinking unit (UVO3 Ltd) and then harvested by

centrifugation at 12,000 *g* for 10 min. Cell pellets were washed with cold 1× PBS and resuspended in 2.4 ml of TN150 buffer (50 mM Tris pH 7.8, 150 mM NaCl, 0.1% NP-40 and 5 mM β-mercaptoethanol) containing protease inhibitors (Complete™, EDTA-free Protease Inhibitor Cocktail) per gram of cells. Suspensions were flash-frozen in droplets and cells were lysed using a Ball Mill MM 400 (five cycles of 3 min at 20 Hz). The mixtures were incubated with 165 units of DNase I (NEB) for 1 h at 25°C and then clarified by centrifugation at 20,000 *g* for 20 min at 4°C.

Protein extracts were subjected to IgG affinity purification on M-280 tosylactivated dynabeads coupled with rabbit IgGs (15 mg of beads per sample). After extensive washes with TN1000 buffer (50 mM Tris pH 7.8, 1 M NaCl, 0.1% NP-40 and 5 mM β-mercaptoethanol), the protein–RNA complexes were eluted by digestion with the TEV protease and treated with 0.2 U of RNase cocktail (RNace-IT, Agilent) to reduce the size of the nascent RNA (note that the 3′ end of nascent transcripts is protected from degradation). The eluates were mixed with guanidine–HCl to a final concentration of 6 M and incubated with Ni-NTA sepharose (Qiagen, 100 μl of slurry per sample) o/n at 4°C. After washing beads, sequencing adaptors were ligated to the RNA molecules as described in the original procedure. Protein–RNA complexes were eluted with 400 μl of elution buffer (50 mM Tris pH 7.8, 50 mM NaCl, 150 mM imidazole, 0.1% NP-40, 5 mM β-mercaptoethanol) and concentrated using Vivacon® ultrafiltration spin columns. Then, proteins were fractionated using a Gel Elution Liquid Fraction Entrapment Electrophoresis (GelFree) system (Expedeon) following manufacturer's specifications and the different fractions were monitored for the presence of Rpb1 by SDS–PAGE. The fractions of interest were treated with 100 μg of proteinase K, and RNAs were purified and reverse-transcribed using reverse transcriptase Superscript IV (Invitrogen).

The cDNAs were amplified by PCR using LA Taq polymerase (Takara), and then, the PCR reactions were treated with 200 U/ml of Exonuclease I (NEB) for 1 h at 37°C. Finally, the DNA was purified using NucleoSpin® columns (Macherey-Nagel) and sequenced on a NextSeq 500 Illumina sequencer.

For Sen1 CRAC, we employed a Sen1-AID strain harbouring a plasmid expressing the pertinent version of Sen1 fused to an HTP-tag at the C terminus (Appendix Table S5). We followed essentially the same protocol as for RNAPII but with several modifications. After cell lysis, the chromatin was solubilized by sonication for 15 min instead of by DNase I treatment. The buffer used for elution from IgG beads was supplemented with NaCl to a final concentration of 0.5 M to reduce unspecific interactions between Sen1 and the IgG beads and we employed a different elution buffer for the Ni-affinity step containing 50 mM Tris–HCl pH 7.5, 50 mM NaCl, 300 mM imidazole, 1% SDS and 1 mM DTT. RNAs were phosphorylated at their 5′ end *in vitro* using the T4 polynucleotide kinase prior to adaptor ligation. Finally, the GelFree fractionation step was omitted because most protein was lost at that step.

### Annotations

For mRNAs, we used the annotations in Challal *et al* (2018). For the 5′ and the 3′ end of mature snoRNAs, we used the annotations in Xu *et al* (2009). In addition, we generated a second annotation were we manually annotated the TTS of snoRNAs according to a genome-

wide mapping of transcripts 3′ ends in a previous report (Roy *et al*, 2016). The criteria we used were to select the earliest 3′ end that is strongly stabilized by deletion of *RRP6*. To facilitate snoRNA analyses, we excluded intronic and polycistronic snoRNAs (Table EV3). For CUTs, we used a subset of 329 out of the 925 originally annotated ones by Xu *et al* (2009) for which we annotated the TTS as for snoRNAs (Table EV4). In order to perform metagene analyses using the mean as the summarizing function, we refined the annotations of both mRNAs and CUTs to exclude those for which a highly expressed gene could be found within a 500 bp window downstream of the annotated TTS in the same strand. The final sets contained 3,393 out of the original 5,752 mRNAs and 321 out of the original CUTs (see Tables EV5–EV8).

### Deep-sequencing dataset processing

RNaseq reads were demultiplexed with bcl2fastq2-2.18.12; adaptor trimming of standard Illumina adaptors was performed with cutadapt 1.15 and reads were subsequently quality-trimmed with trimmomatic (Bolger *et al*, 2014) and mapped to the R64 genome (Cherry *et al*, 2012) with bowtie2 using the default options (Langmead & Salzberg, 2012). Coverage files were normalized by the library size to $10^7$ reads.

CRAC reads were demultiplexed using the pyBarcodeFilter script from the pyCRACutility suite (Webb *et al*, 2014). Next, the 5′ adaptor was clipped with Cutadapt and the resulting insert quality-trimmed from the 3′ end using Trimmomatic rolling mean clipping (Bolger *et al*, 2014). We used the pyCRAC script pyFastqDuplicateRemover to collapse PCR duplicates using a six-nucleotide random tag included in the 3′ adaptor. The resulting sequences were reverse complemented with the Fastx reverse complement that is part of the fastx toolkit (http://hannonlab.cshl.edu/fastx_toolkit/) and mapped to the R64 genome with bowtie2 using "-N 1"option. In the case of RNAPII CRAC, coverage files were normalized by the total number of 3′ end counts to $10^7$ counts. In the case of Sen1 CRAC, the coverage files were normalized by the total number of deletions and set to $10^7$ counts. In those experiments, we had included a spike in consisting in 0.2% of cells of *S. pombe* expressing Rpb1-HTP. Normalization using this spike in provided very similar results but a slightly higher fluctuation between the biological replicates of the same strain, therefore we preferred to use the normalization by total counts.

### Bioinformatic analyses

Bioinformatic analyses were mainly performed using the Galaxy framework (http://galaxy.sb-roscoff.fr and http://deeptools.i.e-freiburg.mpg.de). For metagene analyses, we used a set of tools from the deepTools2 package (Ramírez *et al*, 2016). Strand-specific coverage bigwig files were used as inputs for the computeMatrix tool together with separate annotations for each strand and for each feature (e.g. mRNAs, CUTs, etc.), using a bin size of 10 and the desired reference point. Matrices constructed that way for each strand were subsequently combined using the rbind option of the computeMatrixOperations tool and used as the input for the plotProfile tool. We typically represented the median instead of the mean values to minimize the bias towards the very highly expressed features. Additional analyses performed using the mean values are included in Appendix Figs S2 and S5. The log2 FC of the RNAPII signal in the

mutants relative to the wt was calculated using the tool bigwigCompare. Heatmaps were obtained using the matrices generated by the computeMatrix tool as the input for the plotHeatmap tool. For metagene analyses of Sen1 occupancy normalized to Rpb1, we divided the signal of Sen1 by the Rpb1 signal over 20-nt windows using the tool bigwigCompare. We obtained almost identical results with 30-nt windows (data not shown). Quantification of the average signal over defined windows for the calculation of the read-through index was performed using the multiBigwigSummary tool from the deepTools2 package using the normalized bigwig files for the different samples as inputs. Termination defects were determined by comparing the distributions of RT index of both replicates for Sen1 wt and setting a threshold of 5% for false discovery rate based on these distributions.

For box-plots, we used the ggplot2 R package and statistical analyses were performed with Rstudio.

## Data availability

The data produced in this study are available in the following databases:

- RNAseq and CRAC data: Gene Expression Omnibus GSE117604 (https://www.ncbi.nlm.nih.gov/geo/query/acc.cgi?acc = GSE117604).
- Structure of the Nrd1CID–Sen1 NIM complex: Protein Data Bank 6GC3 (https://www.ebi.ac.uk/pdbe/entry/pdb/6gc3).

**Expanded View** for this article is available online.

### Acknowledgements

We thank M. J. Martin-Niclos for technical assistance and other members of D. L. lab for fruitful discussions. We thank Mara Barucco for her help with bioinformatic analyses. We thank the Roscoff Bioinformatics platform ABiMS (http://abims.sb-roscoff.fr) for providing computational resources and support. This work has benefited from the facilities and expertise of the high-throughput sequencing core facility of I2BC (http://www.i2bc.paris-saclay.fr/). Z.H. was supported by PhD fellowships from the China Scholarship Council and La Ligue contre le Cancer and by a post-doctoral fellowship from the labex "Who am I?". D.L. was supported by the CNRS and the Agence National pour la Recherche grant ANR-12-BSV8-0014-01 and funding under the labex "Who am I" program: ANR-11-IDEX-0005-02 and ANR-11-LABX-0071. O.P was supported by the CNRS and the Agence National pour la Recherche (ANR-16-CE12-0001-01). O.J. was supported by 2017 FEBS Long-Term Fellowship. R.S. was supported by the Czech Science Foundation (GA18-11397S) and Ministry of Education, Youths and Sports of the Czech Republic (CEITEC 2020 project LQ1601). This project has received funding from the European Research Council (ERC) under the European Union's Horizon 2020 research and innovation programme (grant agreement No. 649030 to R.S.). This publication reflects only the author's view and the Research Executive Agency is not responsible for any use that may be made of the information it contains.

### Author contributions

ZH performed biochemical and genetic experiments and prepared RNA samples for deep sequencing. NH and AT performed biochemical experiments. OJ designed experiments, prepared protein samples, performed FA measurements and analysis, analysed NMR spectra, performed structure

calculation; KK collected and processed NMR spectra and assisted with structure calculation; RS designed experiments and assisted with structure calculation. DL advised on experimental design and contributed to manuscript writing. OP conceived the project and performed biochemical experiments and bioinformatic analyses. OP wrote the manuscript with input from all authors.

## Conflict of interest

The authors declare that they have no conflict of interest.

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
