## [Review Process File · The EMBO Journal]

Termination of non-coding transcription in yeast relies on both an RNA Pol II CTD-interaction domain as well as a CTD-mimicking region in Sen1

Zhong Han, Olga Jasnovidova, Nouhou Haidara, Agnieszka Tudek, Karel Kubicek, Domenico Libri, Richard Stefl and Odil Porrua

Review timeline:

Submission date:	17 th Jan 19
Editorial Decision:	14 th March 2019
Revision received:	11 th Sept 2019
Editorial Decision:	7 th Oct 2019
Revision received:	20 th Dec 2019
Editorial Decision:	13 th Jan 2020
Revision received:	23 rd Jan 2020
Accepted:	31 st Jan 2020

Editor: Stefanie Boehm

Transaction Report:

1st Editorial Decision

14th March 2019

Thank you for submitting your manuscript for consideration by The EMBO Journal, as well as providing the pre-decision point-by-point response to the initial reviewer comments. I have now also received the referee's comments on the latter, which I have also discussed with other members of the editorial team.

Overall the reviewers continue to express interest in the study, and find that the proposed experiments will address some, but not all major issues. In particular, the concern regarding experimental data to support the proposed model in which (only) Ser5P of RNAPII recruits Sen1 via its N-terminus, remains (Fig. 7). Specifically, the referee has pointed out that the reduced interaction of Sen1-deltaNter with S5P-CTD (Fig. 6B) observed by co-IP does not unambiguously demonstrate that this is a requirement for recruitment, other mechanisms could exist (i.e. defective Cdk7 recruitment). Furthermore, the referees see a strong need for functional experiments to support the proposed involvement of Ser5P in Sen1 recruitment.

Thus, we find that we can only consider the study further for publication if, in addition to the experiments outlined in the point-by-point response, this concern is also sufficiently addressed. We would therefore require functional experimental proof that supports the proposed requirement of Ser5P in Sen1 recruitment, for example through genetic analysis. (The referee has suggested using strains in which less heptads are mutated, or mutants of Cdk7 to circumvent the CTD mutant strain viability issues.)

If you can fully address the concerns raised by all three referees, including the functional aspect mentioned above, then I would like to invite you to prepare and submit a revised manuscript.

REFeree REPORTS

Referee #1:

The manuscript by Han et al dissects the role of the N and C terminal domains of Sen1. They found that the N-terminal domain binds to P-Ser5 CTD while a short peptide in the N-terminal domain binds to the CID of Nrd1. Interestingly, they found that while NNS termination is largely insensitive to the disruption of the Sen1-Nrd1 interaction, it is heavily dependent on the N-terminal domain of Sen1. The authors argue that the N-terminal domain of Sen1 is the main determinant of Sen1 recruitment, while the Sen1-Nrd1 interaction only contributes marginally. This represents a significant change in the way we usually think of the functions of these factors. In the currently accepted model, Nrd1 (and Nab3) mainly serves as a recruiting platform to Sen1. In the proposed model, Sen1 is directly recruited to the CTD so that Nrd1/Nab3 has to have different functions. The paper is very clear and the data very convincing. The only piece of data that is critically missing is an experiment directly looking at the recruitment of Sen1 in the presence of the N-terminal truncation. The authors mention in their Discussion that this experiment is not possible for technical reasons but it is not clear how. They seem to say that the truncation does not IP to the same efficiency as the WT but details are missing. Was it using tagged proteins? Which tag was used? At which extremity? Given the importance of "recruitment" in this story, it seems like recruitment should be tested rather than just inferred from pulldown assays. One may be able to get around this problem by using less "severe" truncations or point mutations that may kill the Sen1-PSer5 interaction without causing IP problems.

A very recent paper reported that a point mutation in the N-terminal domain of Sen1 and proposed to mediate Sen1-P-Ser2-CTD interactions does not affect Sen1 recruitment (PMID 30639244). The authors should at least comment on this data. Does this mutation affect the P-Ser5-Sen1 interaction described here? It would be interesting to see how this mutant compares to their N-terminal truncation mutant in their assays. As a side note, this same study also provides *in vivo* evidence for the importance of pausing for NNS termination. This study should probably be cited when discussing the importance of pausing with regards to the low translocation processivity of Sen1. Assuming recruitment can be shown directly, the work presented in this manuscript would constitute an important leap forward in our understanding of termination by the NNS pathway.

Referee #2:

In this article, Han et al describe the role of various domains of the Sen1 Helicase, important for termination of ncRNAs, in their interactions the NNS complex and RNA polymerase II carboxy-terminal domain (CTD). First a detailed investigation indicates the role of the C-terminal domain of Sen1, and more specifically through of a new NNS-interacting motif (NIM), in mediating interaction with Nrd1 and Nab3. However, depletion of this C-terminal domain does not result in termination defect neither in a growth phenotype. Second, N-terminal mutants of Sen1 (sen1 Δ Nter) are shown to display a lethal phenotype associated with moderate to strong putative termination defects. The authors also propose that this phenotype and termination defects are at least partially compensated with over-expression of sen1 Δ Nter, suggesting a recruitment defect of sen1 Δ Nter. A deletion of the NIM further aggravates the sen1 Δ Nter phenotype while *in vitro* experiments show possible intra-molecular interactions between N and C-terminal domains of Sen1. Finally, co-IP experiments indicate that depletion of Sen1 Nter does not impair RNAPII recruitment but does specifically decrease Sen1 association to CTD through Ser5-P but not Ser2-P modifications, in contrast to previous reports.

This article presents interesting observations that might be of interest for the specialized field. Co-immunoprecipitation experiments are well performed and highlight the importance of the Nt of Sen1. However and overall, it falls short of presenting many new mechanistic insights to best understand the mechanism at play in the NNS complex during ncRNAs termination. The importance of the Sen1 C-terminal domain seems rather poor for the protein's function and weakly conserved in evolution besides yeast. By contrast, the N-terminal domain is shown to be essential through interaction with RNAPII. But the specificity of the association with Ser5-P presented here is

problematic given a previous report of association with Ser2-P (PMID:22286094) with this domain and a more recent describing involvement of Tyr1-P but not Ser5-P in NNS termination pathway (PMID: 30639244). In my opinion, this last point represents is a major problem in the article and does not allow to draw the model proposed by the authors in Figure 7.

Major points

1- As mentioned above, one of the important observations of the article is the Sen1 N-terminal mutant phenotype, showing a decreased interaction with Ser5-P but not with Rpb1 nor with Ser2-P. This is in contrast to recent report describing a prominent Tyr1, but not Ser5, involvement for NNS function in termination (PMID: 30639244). This data is in apparent conflict with the observation that Ser5 would be required for Sen1. Given the model proposed in the article (Figure 7), it would be a major improvement to strengthen the CTD analyses with Co-IP probed with other RNAPII isoforms, to the least Tyr1-P but also with Thr4-P and Ser7-P. Furthermore, genetic interactions of sen1 mutants with CTD mutants such as S5A, S2A and Y1F could also give indications of specific involvement of the CTD code required in NNS. Finally, how would compare termination defects of the CTD mutants with NNS mutants (part of this information is however described in recent literature).

2- It is not clear why the authors use two different genetic systems for studying the Sen1 N-ter mutants. In Figure 4B, an AID system is described to analyze the termination defect (in the presence of wt Rrp6, if I understand well). Then, Figure 5 presents genome-wide defects of the sen1 Nter mutants in a context where Sen1 is overexpressed and in a Δ rrp6 background. Then again in Figure 6A and 6B the two different strains were used for Co-IP of Sen1 mutants with RNAPII. For the sake of clarity and interpretability of the experiments, it would probably better to present the results for one unique background, preferably with the AID since the phenotype of the GAL1 promoter dependent expression partially rescues the phenotype.

3- There is inconsistencies of the genome-wide profiles showed in Figure 3BC and Figure 5A and E. The wt profiles appear different in shape and median values, which is even more pronounced for the snoRNA profiles. A double peak is observed in snoRNAs profile in Figure 3 over the gene body that is not visible in Figure while the 2 strains should be identical. Figure 5 shows apparently more noise possibly showing lower coverage of sequencing. Finally, no statistical assessment is made for the differences of profiles observed in Figure 5.

4- Northern blots and RNA-seq analyses are based on total RNA experiments (ribo depleted RNA is used for RNA-seq library prep) but no nascent transcription assay is performed in the manuscript to discriminate between stabilization of observed read-through RNAs and transcriptional termination defects (especially given that some of these experiments are performed in the context of an exosome mutant). This should be clearly stated and the reference to 'termination defects' should be amended by this restriction.

Minor point

1- Quantification of the Northern blots of matured vs read-through transcripts would help having a better view of the read-through levels in the different backgrounds.

Referee #3:

In this study, Han and colleagues present a biochemical dissection of *S. cerevisiae* (Sc) Sen1, a conserved DNA/RNA helicase involved in transcription termination of selected RNAP2 genes in complex with Nrd1 and Nab3 RNA-binding proteins (NNS complex), most notably ncRNAs. Previous studies from this group and other labs had cumulated data supporting a model in which Sc Sen1 is recruited to ncRNA genes via interactions with the Nrd1-Nab3 heterodimer. Here, a region located in the C-terminal domain of Sen1 was found to be important for association with the Nrd1-Nab3 complex. This region resembles the Nrd1-interaction motifs (NIM) previously identified by this group for the interaction between Trf4 and Nrd1. Surprisingly, a version of Sen1 deleted for the NIM domain shows normal 3' end processing for snoRNAs and CUTs, suggesting that Sen1 does not require association with Nrd1 for initial recruitment to ncRNA genes. The study also reports that the N-terminal domain of Sen1 is essential for Sen1 function in ncRNA 3' end maturation and important for efficient copurification with Ser5-phosphorylated form of the Rpb1 CTD.

Globally, this manuscript is well written and generally presents high-quality data. Notably, the idea that Sen1 can be recruited to ncRNA genes independently of Nrd1-Nab3 has potentially important repercussions in the view that ncRNA transcription termination is viewed in budding yeast. On the other hand, although this study mainly concerns transcription termination, it lacks evidence of direct measurements of effects on transcription or gene recruitment. Furthermore, as ncRNA transcription termination by NNS appears exclusive to *Saccharomyces* species, it lacks the impact of general interest for a wider audience in the field of transcription termination.

MAJOR COMMENTS:

1. In this study, all of the inferred transcription termination defects were associated with RNA analysis (RNA-seq, Northern blots, RT-qPCR). Although, RNA outputs can reflect consequences on the transcription status, direct measurements of transcription by ChIP or CRAC of RNAP2 need to support the presence and absence of transcription termination defects in this study. This is especially important for the NIM and Nter mutants of Sen1. For instance, the absence of read-through products in the NIM mutants could be due to a redundant nuclease activity despite transcription termination defects.

2. The unambiguously conclude that the association between Sen1 and Nrd1 is not important for recruitment to ncRNA genes, the authors need to present a global analysis comparing the recruitment of wild-type, NIM mutant, and delta-Cterm versions of Sen1 by ChIP-seq. This would provide compelling evidence that Sen1 does not need Nrd1 for recruitment to ncRNA genes.

3. In Figure 5, the authors use an overexpression system using the GAL promoter to conclude that the N-terminal and NIM domains of Sen1 have redundant functions in ncRNA 3' end processing, as demonstrated by RNA-seq. The interpretation for this mainly relies on the slightly greater levels of read-through RNA in the double mutant relative to the single N-ter mutant. Importantly, if the protein levels between the single and double mutant are different (ie. Lower expression of the double mutant relative to single Sen1 mutant), similar results would be observed without

MINOR COMMENTS:

1. P. 6, lines 1-2: to show that Nab3 copurifies with Sen1 via Nrd1 requires further evidence, such as *in vitro* binding assays.

2. Although the authors show IAA-dependent lethality of their Sen1-tagged AID strains, it would be nice to validate efficient protein turnover. How long does it take for endogenous Sen1 to disappear?

1st Revision - authors' response

11th Sept 2019

Response to referees:

We would like to thank all three referees for their thorough examination of our article and their constructive criticism that have allowed us to substantially increase the quality of our manuscript. Following the referees recommendations we have included a good amount of new data, including high resolution genome-wide maps of transcribing RNAPII and Sen1 binding, to directly assess the effect of different Sen1 mutations on transcription and Sen1 recruitment to transcripts. In view of our new results and taking into account some referees suggestions, we have revised our original model to propose a new one that better explains our present data as well as former pieces of evidence. Our responses to specific referees comments are included here below.

Referee #1 (Report for Author)

The manuscript by Han et al dissects the role of the N and C terminal domains of

Sen1. They found that the N-terminal domain binds to P-Ser5 CTD while a short peptide in the N-terminal domain binds to the CID of Nrd1. Interestingly, they found that while NNS termination is largely insensitive to the disruption of the Sen1-Nrd1 interaction, it is heavily dependent on the N-terminal domain of Sen1. The authors argue that the N-terminal domain of Sen1 is the main determinant of Sen1 recruitment, while the Sen1-Nrd1 interaction only contributes marginally. This represents a significant change in the way we usually think of the functions of these factors. In the currently accepted model, Nrd1 (and Nab3) mainly serves as a recruiting platform to Sen1. In the proposed model, Sen1 is directly recruited to the CTD so that Nrd1/Nab3 has to have different functions.

The paper is very clear and the data very convincing. The only piece of data that is critically missing is an experiment directly looking at the recruitment of Sen1 in the presence of the N-terminal truncation. The authors mention in their Discussion that this experiment is not possible for technical reasons but it is not clear how. They seem to say that the truncation does not IP to the same efficiency as the WT but details are missing. Was it using tagged proteins? Which tag was used? At which extremity? Given the importance of "recruitment" in this story, it seems like recruitment should be tested rather than just inferred from pulldown assays. One may be able to get around this problem by using less "severe" truncations or point mutations that may kill the Sen1-PSer5 interaction without causing IP problems.

Response 1: We agree that an important experiment that was missing is the direct assessment of the recruitment of the different Sen1 mutants. In the revised version of the paper we have included high-resolution transcriptome-wide maps of Sen1 binding sites for the wt and mutant variants of Sen1 (figure 6). The HTP-tag that we used in these experiments did not lead to differences in the purification of the different versions of Sen1 (figure S3), allowing us to accurately compare the recruitment of the wt and mutant proteins. Our results indicate that, whereas the interaction of Sen1 with Nrd1 via the NIM contributes moderately to the association of Sen1 with ncRNAs, deletion of the Nter does not decrease Sen1 recruitment. This finding has prompted us to revisit our model. Considering former experimental evidence showing the importance of polymerase pausing for Sen1-mediated termination both *in vitro* (PMID:23748379 and PMID:28180347) and *in vivo* (PMID: 23177741, PMID 30639244) and the fact that the S5P mark is the most prominent one at promoter-proximally paused RNAPII (PMID:27288397), in the revised version of our manuscript we propose that the interaction of Sen1 Nter with the S5P-CTD is critical to couple Sen1 translocation to RNAPII pausing to enable efficient termination.

A very recent paper reported that a point mutation in the N-terminal domain of Sen1 and proposed to mediate Sen1-P-Ser2-CTD interactions does not affect Sen1 recruitment (PMID 30639244). The authors should at least comment on this data. Does this mutation affect the P-Ser5-Sen1 interaction described here? It would be interesting to see how this mutant compares to their N-terminal truncation mutant in their assays. As a side note, this same study also provides *in vivo* evidence for the importance of pausing for NNS termination. This study should probably be cited when discussing the importance of pausing with regards to the low translocation processivity of Sen1. Assuming recruitment can be shown directly,

the work presented in this manuscript would constitute an important leap forward in our understanding of termination by the NNS pathway.

Response 2: In the revised version of our article we have discussed the data in the mentioned paper (PMID 30639244). Indeed, their finding that the R302W mutation proposed to mediate the interaction of Sen1 with the S2P-CTD in PMID:22286094 does not affect Sen1 recruitment is consistent with our results showing that deletion of the whole Nter does not impair Sen1 recruitment. In addition, the authors of the report PMID 30639244 did not detect any significant termination defect on the sen1R302W mutant, which is in agreement with our unpublished results showing the absence of any growth or transcription termination phenotype in this mutant. Therefore, we believe that exploring in more detail the behavior of the sen1R302W mutant would likely not help understanding the role of the Sen1-RNAPII CTD interaction in NNS-dependent termination.

Referee #2 (Report for Author)

In this article, Han et al describe the role of various domains of the Sen1 Helicase, important for termination of ncRNAs, in their interactions the NNS complex and RNA polymerase II carboxy-terminal domain (CTD). First a detailed investigation indicates the role of the C-terminal domain of Sen1, and more specifically through of a new NNS-interacting motif (NIM), in mediating interaction with Nrd1 and Nab3. However, depletion of this C-terminal domain does not result in termination defect neither in a growth phenotype. Second, N-terminal mutants of Sen1 (sen1 Δ Nter) are shown to display a lethal phenotype associated with moderate to strong putative termination defects. The authors also propose that this phenotype and termination defects are at least partially compensated with over-expression of sen1 Δ Nter, suggesting a recruitment defect of sen1 Δ Nter. A deletion of the NIM further aggravates the sen1 Δ Nter phenotype while in vitro experiments show possible intra-molecular interactions between N and C-terminal domains of Sen1. Finally, co-IP experiments indicate that depletion of Sen1 Nter does not impair RNAPII recruitment but does specifically decrease Sen1 association to CTD through Ser5-P but not Ser2-P modifications, in contrast to previous reports.

This article presents interesting observations that might be of interest for the specialized field. Co-immunoprecipitation experiments are well performed and highlight the importance of the Nt of Sen1. However and overall, it falls short of presenting many new mechanistic insights to best understand the mechanism at play in the NNS complex during ncRNAs termination.

Response 3: In this article we address the mechanisms that ensure timely recruitment and function of Sen1 at ncRNAs, which is a question of major biological relevance. Before our present work Sen1 was supposed to be recruited by its partners in the NNS-complex, Nrd1 and Nab3, although the molecular details of the interaction between Sen1 and Nrd1 and Nab3 remained unknown. In this study we have identified the protein regions involved in the association of Sen1 with its partners and we have provided the structural details of these interactions. We have shown that, contrary to the current model, this interaction plays an accessory role in

transcription termination. In addition, we have shown that non-coding transcription termination critically depends on the interaction of Sen1 to the Ser5P CTD via its N-terminal domain. Finally, we have generated high-resolution RNA binding maps for Sen1 of very high quality that have allowed us to characterize the genomic distribution of Sen1 and assess directly the role of different protein interactions in Sen1 recruitment. We believe that our study represents very significant progress beyond the state of the art, an opinion that is shared by the two other referees.

The importance of the Sen1 C-terminal domain seems rather poor for the protein's function and weakly conserved in evolution besides yeast. By contrast, the N-terminal domain is shown to be essential through interaction with RNAPII. But the specificity of the association with Ser5-P presented here is problematic given a previous report of association with Ser2-P (PMID:22286094) with this domain and a more recent describing involvement of Tyr1-P but not Ser5-P in NNS termination pathway (PMID: 30639244). In my opinion, this last point represents is a major problem in the article and does not allow to draw the model proposed by the authors in Figure 7.

Response 4: As pointed above (see response 2) we have some reasonable doubts about the validity of the data presented in article PMID:22286094 regarding the role of the interaction between Sen1 and the S2P-CTD. In fact, the data presented in paper PMID: 30639244 together with the results that we present in our revised manuscript do not support the idea that the interaction of Sen1 with the S2P-CTD via the R302 residue is necessary for the recruitment of Sen1 to nascent transcripts and for transcription termination. Because there is no structure available for the Nter of Sen1 in complex with the CTD, our present results do not allow us to exclude that the Nter can interact both with S2P-CTD and with S5P-CTD. This is now clearly stated in the revised manuscript. However, our results showing that the S5P-CTD interacting domain of Nrd1 (the CID) can functionally replace the Nter of Sen1 is a very strong argument supporting the idea that the essential function of the Nter is to recognize the S5P-CTD. This notion is now further strengthened by our new genetic and biochemical data showing that the CID of Pcf11, which recognizes preferentially S2P, cannot support cell growth and efficient transcription termination when located at the place of the Nter (see new figure 5).

Concerning the apparent discrepancy with article PMID: 30639244, from our point of view, our results are not incompatible with the data in this paper (see also response 1). First, in that article they provide evidence that promoter-proximal pausing depends on Y1 phosphorylation and is critical for efficient termination by the NNS-complex. The idea that termination requires polymerase pausing is fully consistent with previous data from ours (PMID:23748379 and PMID:28180347) and from another group (PMID: 23177741). In the light of our new results showing that Sen1 Nter is not required for Sen1 recruitment to ncRNAs (see new figure 6), we have revised our model (see new figure 7 and new discussion) to propose that the interaction of Sen1 Nter with the S5P-CTD is important for the kinetic coupling between Sen1 translocation and RNAPII pausing during termination.

Second, the authors of PMID: 30639244 conclude that the S5 phosphorylation is not important for NNS-dependent termination based on their finding that a S5A mutant version of RNAPII does not display any clear transcription termination defects. However, for reasons that are not completely clear, this mutation induces a

massive increase in promoter-proximal polymerase pausing. According to the current model of kinetic competition between termination and elongation, this enhancement of pausing would favor termination by Sen1, thus compensating for the lack of interaction of Sen1 with the S5P-CTD. These aspects are discussed in the revised manuscript.

Major points

1- As mentioned above, one of the important observations of the article is the Sen1 N-terminal mutant phenotype, showing a decreased interaction with Ser5-P but not with Rpb1 nor with Ser2-P. This is in contrast to recent report describing a prominent Tyr1, but not Ser5, involvement for NNS function in termination (PMID: 30639244). This data is in apparent conflict with the observation that Ser5 would be required for Sen1.

See response 4.

Given the model proposed in the article (Figure 7), it would be a major improvement to strengthen the CTD analyses with Co-IP probed with other RNAPII isoforms, to the least Tyr1-P but also with Thr4-P and Ser7-P.

Response 5: In the revised version we have included a new figure S2 where we have assessed the impact of the Nter deletion in the interaction of Sen1 with S7P-CTD and with the unphosphorylated CTD, which is the CTD form that is preferentially recognized by the antibody 8WG16 (abcam). We have not observed any decrease in the association of Sen1 Δ Nter with those forms of the CTD. We could not detect Y1P using the commercial antibody 3D12 (Millipore), but at this point we do not know if this is due to a problem with the batch of antibody used or to rapid dephosphorylation of Y1P-CTD in our extracts for unknown reasons. However, in the mentioned paper (PMID: 30639244), the authors do not observe any difference in the genomic distribution of Sen1 in the Y1F and they actually provide evidence supporting a role of Y1P in mediating promoter-proximal pausing to facilitate NNS-dependent termination, rather in direct binding to the NNS-complex. We did not test a possible role for Sen1 Nter in mediating the recognition of T4P because this possibility is not supported by any data in the literature.

Furthermore, genetic interactions of sen1 mutants with CTD mutants such as S5A, S2A and Y1F could also give indications of specific involvement of the CTD code required in NNS. Finally, how would compare termination defects of the CTD mutants with NNS mutants (part of this information is however described in recent literature).

Response 6: Most CTD mutants are very sick or even lethal, likely because of the alteration of the association of multiple factors (not only termination factors) with the CTD. Therefore, testing the genetic interactions with NNS-mutants is not always possible and, when feasible, might provide results that are very difficult to interpret and unlikely to provide much mechanistic information. We tried to perform some genetic analyses with a thermosensitive mutant of Kin28, the major S5 kinase, however the frequent suppressors obtained with that strain during the

experiment did not allow us to obtain conclusive results. Instead, in order to further support the notion that the interaction of Sen1 Nter with S5P and not S2P is critical for transcription termination at non-coding genes, we have performed the aforementioned experiments with the chimeric constructs harbouring either Nrd1 or Pcf11 CID fused to sen1 Δ Nter (see response 4 and new figure 5).

Regarding the comparison of the termination defects of the CTD mutants with NNS mutants, only the data generated in the same genetic context and growth conditions, using the same technique (ChIP-chip, RNAseq, etc) can be directly compared. There are not many data of this kind available and it is out of the scope of this paper to repeat the experiments previously performed in the CTD mutants in parallel with the Sen1 mutants characterized in our study.

2- It is not clear why the authors use two different genetic systems for studying the Sen1 N-ter mutants. In Figure 4B, an AID system is described to analyze the termination defect (in the presence of wt Rrp6, if I understand well). Then, Figure 5 presents genome-wide defects of the sen1 Nter mutants in a context where Sen1 is overexpressed and in a Δ rrp6 background. Then again in Figure 6A and 6B the two different strains were used for Co-IP of Sen1 mutants with RNAPII. For the sake of clarity and interpretability of the experiments, it would probably better to present the results for one unique background, preferably with the AID since the phenotype of the GAL1 promoter dependent expression partially rescues the phenotype.

Response 7: In the revised version of our manuscript we have used only one genetic system, the Sen1-AID system, to assess the role of the different mutations on transcription termination (see new figures 3 and 4). Regarding the CoIPs, for technical reasons we are obliged to use the strains that overexpress Sen1 from the GAL1 promoter when we use Sen1 as a bait to detect the interaction with the different RNAPII phospho-isoforms. Because RNAPII is a very abundant protein, contrary to Sen1, we obtain very high levels of background binding of RNAPII to beads that preclude accurate assessment of the interaction between Sen1 and the polymerase. In fact, the previous report on the role of Sen1 Nter on the interaction with RNAPII (PMID:22286094) lacked the appropriate controls (e.g. a coIP with untagged Sen1) to account for background levels of RNAPII and evaluate this interaction. This issue is resolved by increasing the levels of Sen1 protein.

3- There is inconsistencies of the genome-wide profiles showed in Figure 3BC and Figure 5A and E. The wt profiles appear different in shape and median values, which is even more pronounced for the snoRNA profiles. A double peak is observed in snoRNAs profile in Figure 3 over the gene body that is not visible in Figure while the 2 strains should be identical. Figure 5 shows apparently more noise possibly showing lower coverage of sequencing. Finally, no statistical assessment is made for the differences of profiles observed in Figure 5.

Response 8: The differences in the profiles in former figures 3 and 5 could be due to many different reasons, none of them invalidating the results: the genetic background is different, cells were grown in different media in both experiments (rich vs minimal medium), the libraries were prepared at different moments and samples were sequenced independently. Nonetheless, we have removed the

experiments in figure 5 (although perfectly consistent with our new data) to include instead high-resolution genome-wide maps of transcribing RNAPII to directly assess the efficiency of transcription termination by the different Sen1 variants (see new figure 4). In order to evaluate the statistical significance of the differences in RNAPII distribution observed in the metagene profiles, we have calculated the readthrough (RT) index for each transcript (see new figure 4 for details) and we have applied a t-test to compare the population of RT index of each mutant with that of the wt for each feature (e.g. CUTs, snoRNAs). We have also quantified and represented the termination defects at each ncRNA individually (see new figures 4 and EV4).

4- Northern blots and RNA-seq analyses are based on total RNA experiments (ribo depleted RNA is used for RNA-seq library prep) but no nascent transcription assay is performed in the manuscript to discriminate between stabilization of observed read-through RNAs and transcriptional termination defects (especially given that some of these experiments are performed in the context of an exosome mutant). This should be clearly stated and the reference to 'termination defects' should be amended by this restriction.

Response 9: As mentioned in **response 8**, in the revised version of our article we have provided new genome-wide data to directly assess the termination defects of the different mutants (new figures 4 and EV4).

Minor point

1- Quantification of the Northern blots of matured vs read-through transcripts would help having a better view of the read-through levels in the different backgrounds.

Response 10: As indicated in **response 8**, in the revised manuscript we have included the appropriate quantifications and statistical assessment of new genome-wide transcription data, which provide more extensive and accurate assessment of the termination defects in the different mutants than quantifications of the different species in our northern blot assays.

Referee #3 (Report for Author)

In this study, Han and colleagues present a biochemical dissection of *S. cerevisiae* (Sc) Sen1, a conserved DNA/RNA helicase involved in transcription termination of selected RNAP2 genes in complex with Nrd1 and Nab3 RNA-binding proteins (NNS complex), most notably ncRNAs. Previous studies from this group and other labs had cumulated data supporting a model in which Sc Sen1 is recruited to ncRNA genes via interactions with the Nrd1-Nab3 heterodimer. Here, a region located in the C-terminal domain of Sen1 was found to be important for association with the Nrd1-Nab3 complex. This region resembles the Nrd1-interaction motifs (NIM) previously identified by this group for the interaction between Trf4 and Nrd1. Surprisingly, a version of Sen1 deleted for the NIM domain shows normal 3'

end processing for snoRNAs and CUTs, suggesting that Sen1 does not require association with Nrd1 for initial recruitment to ncRNA genes. The study also reports that the N-terminal domain of Sen1 is essential for Sen1 function in ncRNA 3' end maturation and important for efficient copurification with Ser5-phosphorylated form of the Rpb1 CTD.

Globally, this manuscript is well written and generally presents high-quality data. Notably, the idea that Sen1 can be recruited to ncRNA genes independently of Nrd1-Nab3 has potentially important repercussions in the view that ncRNA transcription termination is viewed in budding yeast. On the other hand, although this study mainly concerns transcription termination, it lacks evidence of direct measurements of effects on transcription or gene recruitment. Furthermore, as ncRNA transcription termination by NNS appears exclusive to *Saccharomyces* species, it lacks the impact of general interest for a wider audience in the field of transcription termination.

Response 11: We appreciate the positive comments of the referee on our study. However, we do not agree with the idea that the fact that the NNS-complex is not universally conserved in eukaryotes renders studies on the function of this complex uninteresting for a wide audience. The reports on the mechanisms of non-coding transcription termination in budding yeast have inspired works on other systems (citations are the proof of that), including the human model. Furthermore, although the precise actors might differ, the general scheme and mechanisms revealed in our study might apply to multitude of other organisms.

MAJOR COMMENTS:

1. In this study, all of the inferred transcription termination defects were associated with RNA analysis (RNA-seq, Northern blots, RT-qPCR). Although, RNA outputs can reflect consequences on the transcription status, direct measurements of transcription by ChIP or CRAC of RNAP2 need to support the presence and absence of transcription termination defects in this study. This is especially important for the NIM and Nter mutants of Sen1. For instance, the absence of read-through products in the NIM mutants could be due to a redundant nuclease activity despite transcription termination defects.

Response 12: As mentioned in **response 8** in the revised manuscript we have included high-resolution maps of transcribing RNAPII generated by CRAC in the different Sen1 mutant contexts. The results obtained with these new datasets support well the original conclusions of our paper, which were: i) deletion of the NIM impairs transcription termination at a small subset of non-coding genes; ii) deletion of the Nter provokes strong and widespread defects in non-coding transcription termination and iii) concomitant deletion of the Nter and the NIM aggravates the termination defects at a fraction of ncRNAs.

2. The unambiguously conclude that the association between Sen1 and Nrd1 is not important for recruitment to ncRNA genes, the authors need to present a global analysis comparing the recruitment of wild-type, NIM mutant, and delta-Cterm

versions of Sen1 by ChIP-seq. This would provide compelling evidence that Sen1 does not need Nrd1 for recruitment to ncRNA genes.

Response 13: As mentioned in **response 1** in the revised version of our article we have directly assessed the recruitment of the different Sen1 mutants to ncRNAs by performing Sen1 CRAC (new figure 6). We have generated high-quality near single-nucleotide resolution datasets showing that: i) the presence of the NIM provides a modest contribution to the recruitment of Sen1 to ncRNAs and ii) deletion of the Nter does not generally decrease Sen1 recruitment to ncRNAs, contrary to our expectations. These observations have led us to modify our original model (see new figure 7 and discussion) to explain the role of the NIM and the Nter in Sen1 recruitment and function.

3. In Figure 5, the authors use an overexpression system using the GAL promoter to conclude that the N-terminal and NIM domains of Sen1 have redundant functions in ncRNA 3' end processing, as demonstrated by RNA-seq. The interpretation for this mainly relies on the slightly greater levels of read-through RNA in the double mutant relative to the single N-ter mutant. Importantly, if the protein levels between the single and double mutant are different (ie. Lower expression of the double mutant relative to single Sen1 mutant), similar results would be observed without

Response 14: As mentioned above, we have excluded the RNA-seq experiments in the overexpression system in the revised article. In the light of our new RNAPII CRAC data (see **responses 1 and 13**) we can confirm that the simultaneous deletion of the NIM and the Nter of Sen1 provokes stronger termination effects than the deletion of the Nter alone at a subset of ncRNAs, although both proteins (single and double mutants) exhibit identical expression levels (see figure S1C). However, our new Sen1 CRAC data do not support the notion of redundant roles for these two protein regions but rather different functions that provoke additive defects in termination when both are impaired.

MINOR COMMENTS:

1. P. 6, lines 1-2: to show that Nab3 copurifies with Sen1 via Nrd1 requires further evidence, such as in vitro binding assays.

Response 15: We believe that the fact that the interaction of Nab3 with Sen1 is lost in the Sen1 mutant lacking the Nrd1-interaction motif is a reasonably strong argument to propose that the interaction between Sen1 and Nab3 is mediated by Nrd1.

2. Although the authors show IAA-dependent lethality of their Sen1-tagged AID strains, it would be nice to validate efficient protein turnover. How long does it take for endogenous Sen1 to disappear?

Response 16: Unfortunately, we have not found an antibody against the AID-tag that is sufficiently sensitive to detect Sen1-AID in protein extracts. Because of its large size and low abundance, Sen1 is very difficult to detect by western blot in whole-cell protein extracts. We have determined the right depletion time by monitoring the termination defects by northern blot upon addition of IAA at various time points.

2nd Editorial Decision

7th Oct 2019

Thank you for submitting your revised manuscript for our consideration. It has now been seen once more by two of the original referees (see comments below). As you will see, both referees unfortunately continue to have several major concerns regarding the model and interpretation of the interaction of Sen1 N-terminus with RNAPII Ser5P. In our previous decision letter, we noted that we could only consider the study further if this issue was sufficiently addressed and we discussed the current reports (in light of the initial decision) again within the editorial team. Given that you have added substantial improvements, in particular the CRAC experiment, which both referees also explicitly acknowledge, we would like to give you the opportunity to address the remaining issues in an exceptional second round of revision. Referee #2's point 3 regarding the CRAC data analysis can be addressed by re-analyzing the already acquired data and replacing plots or providing them in addition. Furthermore, as both referees are not convinced by the interpretation of the model, providing detailed arguments and suggesting clarifying experiments, these (or equivalent alternative) experiments should at least be attempted and the resulting data provided in the next response to the referees. In addition, these results and the referees' arguments have to be taken into account when proposing and discussing an overall model revised to reflect the remaining additional interpretation possibilities regarding RNAPII Ser5P. As indicated above, it is normally EMBO Journal's policy to allow only one round of major revision, such that it is now crucial that you address the referee's concerns fully - experimentally, through data reanalysis, as well as critical revision of the model proposed. If you have any questions regarding this revision or would like to discuss how to proceed in more detail, please feel free to contact me.

REFeree REPORTS

Referee #1:

The revised manuscript by Han et al is significantly improved over the original submission. The most significant addition consists of CRAC datasets that 1) allowed to more directly show termination defects (compared to initial RNA-seq data) and 2) allowed to look at recruitment defects of the different Sen1 mutants. Notably, these new data forced the authors to revise their initial model. Indeed, the new data clearly show that the recruitment of the Sen1DNter mutant is not impaired, contrarily to what they had originally assumed. Overall, the authors have adequately addressed the criticisms and the revised manuscript is clear and solid.

My only problem is with their final model, which I think does not fit well with the overall knowledge. At this stage, too many grey zones prevent the elaboration of a single model. The proposed model suggests that the interaction of Sen1-Nter with P-Ser5 (which occurs post Sen1 recruitment) somehow enables its translocation along the RNA. This model is difficult to reconcile with the fact that the author's own *in vitro* data (Han et al., 2017) showed no effect of truncating the N-ter of Sen1 on translocation. Most importantly, the model is difficult to reconcile with the fact that S5A mutants have very mild NNS termination defects (Collin et al, 2019) compared to those shown here with the DNter mutant by CRAC. On page 18, the authors propose that this may be explained by a "major pausing" occurring in the S5A mutant that would bypass the requirement of the Sen1-N-PSer5 interaction. This would be a reasonable explanation should the S5A elicit pausing defects, but it has been shown that the accumulation of Pol II in the 5' region of genes in S5A (and kin28 mutants) is not due to increased pausing but rather to slower promoter release. The increased signal of Pol II in S5A hence reflects a more stable (long-lived) PIC. This is well described in papers

from the Struhl and Robert labs (24746699, 27773675, 24704787, 2777367). Unless the author would argue that Sen1 can join Pol II as part of the PIC, the explanation does not hold. At the very least, this section of the discussion should be revised to accurately reflect the S5A defect. The authors should also consider models where the Sen1-Nter mediates its effect independently of P-Ser5, or perhaps with alternative (and perhaps functionally redundant) phosphorylation states.

Referee #2:

Major points previously addressed and discussed by the authors:

1- The co-IP controls with other RNAPII isoforms required were not provided. That the other isoforms (Ser7P, Thr4P, Tyr1P) are shown in this experiment seems rather important to assess the specificity of Ser5P co-IP signal. And yes, there is evidence that Thr4P is likely involved in termination of snoRNA termination (PMIDs: 28465432 and 30598543). The sup fig2 mentioned in the 'Response 5' (containing Ser7P and 8WG16 co-IP) by the authors is not present in the revised manuscript (maybe a simple mistake?).

If genetic interaction cannot be directly tested with kin28 ts nor CTD mutants, kin28 altered specificity mutants could be used to address the question of importance of Ser5P by TFIIH. Another option to address the question of Ser5P and Sen1 interaction would be to show that in the context RNAPII CTD isoforms co-IP with Sen1 (Fig. 6B), a recovery of Ser5P signal can be obtained when the Nrd1 CID is fused to Sen1 (the mutant is shown in Fig. 6C) as compared to the Δ Nter situation. In the present form, I remain not completely convinced by the evidence provided that Ser5P is important and/or specific for the event of termination involving Sen1.

2- The answer provided is appropriate.

3- The novel data (CRAC) provided indeed brings an added value to the manuscript by indicating that a termination defect most likely causes the observed phenotype. But as previously, it is likely that noise is added by the way the data is processed. One of the difficulties with the yeast genome analyses is its compactness which means genes are very often spaced by less than a few hundred bp. Also, when performing genic-basic metaprofile analyses, it is best to remove from the data those genes that are not expressed/not bound in the ref data set (WT here). Typically starting from roughly 6000 mRNA coding genes and restricting RNAPII binding level a few hundred genes are left for a metaprofile analyses. Similarly, CUTs and snoRNAs will be reduced to a lower number but providing overall a more robust, less diluted by noise or mistaken by surrounding region signals. Examples of selections for metagene analyses can be found in PMID 30639244 and 19450536. Furthermore, it is mentioned in the methods that median profiles for CRAC were performed rather than average. None of these methods is indeed perfect since average profiles can overestimate a subset of highly contributing signals (as mentioned by the authors) and the median can largely ignore an important part of the data. In both cases, bias can be introduced depending on the nature of the signal distribution. However, using averages profiles of several quartiles can help checking the homogeneity of distribution of the signals. Such presentation would improve the accuracy of the analysis.

4- The question was solved by the novel CRAC experiment.

2nd Revision - authors' response

20th Dec 2019

Response to referees:

Referee #1:

The revise manuscript by Han et al is significantly improved over the original submission. The most significant addition consists of CRAC datasets that 1) allowed to more directly show termination defects (compared to initial RNA-seq data) and 2) allowed to look at recruitment defects of the different Sen1 mutants. Notably, these new data forced the authors to revise their initial model. Indeed, the new data clearly show that the recruitment of the Sen1 Δ Nter mutant is not impaired, contrarily to what they had originally assumed. Overall, the authors have adequately addressed the criticisms and the revised manuscript is clear and solid.

My only problem is with their final model, which I think does not fit well with the overall knowledge. At this stage, too many grey zones prevent the elaboration of a single model. The proposed model suggests that the interaction of Sen1-Nter with P-Ser5 (which occurs post Sen1 recruitment) somehow enables its translocation along the RNA. This model is difficult to reconcile with the fact that the author's own in vitro data (Han et al., 2017) showed no effect of truncating the N-ter of Sen1 on translocation.

Response 1: We thank the referee for his/her comment and we realize that in the previous version of the manuscript the model regarding the role of the Nter in termination was not sufficiently clear, we apologize for this. In the new revised version we have tried to further clarify this aspect by re-writing part of the discussion and by adding new data. In fact, we did not intend to propose that the Nter enhances Sen1 translocation along the RNA (i.e. translocation rate and/or processivity), as we have not observed this upon deletion of the Nter *in vitro* (Han et al., 2017). We rather propose that the interaction of Sen1 Nter with the CTD of RNAPII could increase the chances that translocating Sen1 makes termination-productive encounters with the paused RNAPII. Because Sen1 has low processivity (i.e. it dissociates frequently during translocation), polymerase pausing creates a window of opportunity for Sen1 to reach RNAPII and induce termination. The persistence of RNAPII at pausing determines the size of the window of opportunity for termination and, the Sen1-CTD interaction could be critical to position Sen1 within this window. We suggest that, in the absence of interaction with the CTD, Sen1 undergoes multiple futile cycles of association-dissociation from the RNA, explaining the termination defects observed in the Δ Nter mutant. We have provided evidence supporting this model in the new figure 7A. Indeed, we have shown that the Nter of Sen1 is not (or much less) required for termination in the presence of the elongation inhibitor 6-azauracile. This compound decreases the intracellular pools of GTP and UTP, leading to an overall decrease in the transcription elongation rate. This is most likely the result of strong and frequent polymerase pausing at each position where the nucleotide to be added is limiting. Therefore, we propose that impairing elongation by RNAPII extends the window of opportunity for Sen1-dependent termination sufficiently to bypass the requirement for the Sen1-CTD interaction.

Most importantly, the model is different to reconcile with the fact that S5A mutants have very mild NNS termination defects (Collin et al, 2019) compared to those shown here with the DNter mutant by CRAC. On page 18, the authors propose that this may be explained by a "major pausing" occurring in the S5A mutant that would bypass the requirement of the Sen1-N-P-Ser5 interaction. This would be a reasonable explanation should the S5A elicit pausing defects, but it has been shown that the accumulation of Pol II in the 5' region of genes in S5A (and kin28 mutants) is not due to increased pausing but rather to slower promoter release. The increased signal of Pol II in S5A hence reflects a more stable (long-lived) PIC. This is well described in papers from the Struhl and Robert labs (24746699, 27773675, 24704787, 2777367). Unless the author would argue that Sen1 can join Pol II as part of the PIC, the explanation does not hold. At the very least, this section of the discussion should be revised to accurately reflect the S5A defect. The authors should also consider models where the Sen1-Nter mediates its effect independently of P-Ser5, or perhaps with alternative (and perhaps functionally redundant) phosphorylation states.

Response 2: We agree with the referee that the current model posits that the S5 phosphorylation is required for promoter release. We have revised the discussion to clearly reflect this. However, we believe that this poses a major problem to accurately determine the role of S5P in NNS-dependent termination. Indeed, as the referee indicates, the polymerases that are retained at the promoter cannot be a substrate for Sen1-mediated termination because Sen1 strictly needs to interact with the nascent RNA and RNAPII needs to reach the termination region to be subjected to termination. Therefore, we have set out to re-analyse the role of S5P for NNS-dependent termination. Specifically, we have performed northern blot analyses in a kin28 ts and a S5A mutant (CAND system, Collin et al 2019) in an appropriate genetic background (Δ rrp6) to detect both the primary products of termination and the readthrough RNAs that result from defective termination (see new appendix figure S6). We would like to emphasize that detecting the primary products of termination is crucial as the termination defects can only be evaluated by comparing the fraction of transcripts terminated at the primary site with the fraction of extended or readthrough transcripts. In the aforementioned study (Collin et al, 2019) the primary products of termination could not be detected because the experiments were not performed in an exosomal-deficient background. Indeed, we have

found that inactivation of kin28 or transient depletion of the wt version of Rpb1 in the S5A CAND system, both lead to a dramatic decrease in the production of RNAPII transcripts (both ncRNAs and an abundant mRNA). The small amount of ncRNAs that could be detected were similar in size to the properly terminated transcripts in the control, but it cannot be ascertained if this is due to either incomplete inactivation of kin28 (or incomplete depletion of the wt Rpb1 version in the CAND system) or to efficient termination in the absence of S5P. Thus, these results are not conclusive regarding the role of S5P in NNS-dependent termination. However, this also indicates that the former data in Collin et al, 2019, actually did not allow to determine the role of S5P because of the effect of the S5A mutation on polymerase promoter release. Nevertheless, in the absence of a formal proof that the S5P is the only phosphomark supporting the interaction with the Nter to favour efficient NNS-dependent termination, and the technical limitations to obtain such proof, we have decided to revise our model to include the possibility that other phosphorylated forms of the CTD can also be bound by Sen1 and function in termination.

Referee #2:

Major points previously addressed and discussed by the authors:

1- The co-IP controls with other RNAPII isoforms required were not provided. That the other isoforms (Ser7P, Thr4P, Tyr1P) are shown in this experiment seems rather important to assess the specificity of Ser5P co-IP signal. And yes, there is evidence that Thr4P is likely involved in termination of snoRNA termination (PMIDs: 28465432 and 30598543). The sup fig2 mentioned in the 'Response 5' (containing Ser7P and 8WG16 co-IP) by the authors is not present in the revised manuscript (maybe a simple mistake?).

Response 3: The co-IP revealed with antibodies against Ser7P and antibody 8WG16 was actually provided in the Appendix figure S2. We have now included the analysis of Thr4P, as requested (please, note that the figure is now renamed as appendix figure S3). We do not observe any decrease in the amount of Thr4P CTD associated with Sen1 upon deletion of the Nter. As mentioned before, the possibility that Sen1 recognizes Tyr1P via the Nter is extremely unlikely considering that the CID of Nrd1 can functionally replace the Nter of Sen1 and that Tyr1P inhibits the interaction of Nrd1 CID with the CTD.

If genetic interaction cannot be directly tested with kin28 ts nor CTD mutants, kin28 altered specificity mutants could be used to address the question of importance of Ser5P by TFIIH. Another option to address the question of Ser5P and Sen1 interaction would be to show that in the context RNAPII CTD isoforms co-IP with Sen1 (Fig. 6B), a recovery of Ser5P signal can be obtained when the Nrd1 CID is fused to Sen1 (the mutant is shown in Fig. 6C) as compared to the Δ Nter situation. In the present form, I remain not completely convinced by the evidence provided that Ser5P is important and/or specific for the event of termination involving Sen1.

Response 4: We have performed additional northern blot assays in a kin28 ts mutant as well as in the S5A mutant reported in PMID 30639244 to assess whether decrease in S5 phosphorylation has a negative effect on Sen1-dependent termination, similar to deletion of the Nter (see **response 2**). As shown in the new appendix figure S6, both inactivation of Kin28 at 37°C and mutation of S5 to A provoke a major decrease in transcription. This supports the current model that S5 phosphorylation is required for promoter escape, but implicates that any experiment preventing S5 phosphorylation would not allow addressing the importance of S5P for Sen1-dependent termination because polymerases that are retained at the promoter cannot be subjected to termination. Indeed, as we showed before, Sen1 strictly requires the interaction with a sufficiently long nascent RNA to induce termination. It is extremely likely that the presence of the Nrd1 CID fused to Sen1 allows the recovery of the interaction between Sen1 and the Ser5P CTD because the capacity of Nrd1CID to recognize the Ser5P CTD is very well established (PMID 18660819 and 22892239, for example), however, this would not prove that this recovery is the reason of the suppression we observe (although it would be the most likely explanation). We believe that the fact that the deletion of the Nter only decreases the association of Sen1 with the S5P CTD (and not with other phosphorylated

forms of the CTD) and that the CID of Nrd1 that recognizes better S5P than other phosphorylations in the CTD are strong arguments to claim that Sen1 Nter has a preference for the S5P CTD. However, because we cannot provide a formal proof that only the S5P can support the interaction of Sen1 with the CTD during termination, we have opted to tune-down the emphasis on the importance of the S5 phosphorylation over other phosphomarks for Sen1-mediated termination and we have included in the discussion and in our revised model (figure 7) the possibility that other phosphoisoforms of the CTD also support the interaction with Sen1 for efficient termination.

2- The answer provided is appropriate.

3- The novel data (CRAC) provided indeed brings an added value to the manuscript by indicating that a termination defect most likely causes the observed phenotype. But as previously, it is likely that noise is added by the way the data is processed. One of the difficulties with the yeast genome analyses is its compactness which means genes are very often spaced by less than a few hundred bp. Also, when performing genic-basic metaprofile analyses, it is best to remove from the data those genes that are not expressed/not bound in the ref data set (WT here). Typically starting from roughly 6000 mRNA coding genes and restricting RNAPII binding level a few hundred genes are left for a metaprofile analyses. Similarly, CUTs and snoRNAs will be reduced to a lower number but providing overall a more robust, less diluted by noise or mistaken by surrounding region signals. Examples of selections for metagene analyses can be found in PMID 30639244 and 19450536. Furthermore, it is mentioned in the methods that median profiles for CRAC were performed rather than average. None of these methods is indeed perfect since average profiles can overestimate a subset of highly contributing signals (as mentioned by the authors) and the median can largely ignore an important part of the data. In both cases, bias can be introduced depending on the nature of the signal distribution. However, using averages profiles of several quartiles can help checking the homogeneity of distribution of the signals. Such presentation would improve the accuracy of the analysis.

Response 5: We understand the concern of the referee about the possible noise added by the signal of surrounding regions and the presence of genes/CUTs with very low expression in the sets we have used.

Regarding the interference of surrounding regions, our CRAC data have the advantage relative to the ChIP data in PMID 30639244 and 19450536 mentioned by the referee that they are strand specific and provide near single nucleotide resolution. Therefore, we do not need to be so restrictive in the choice of genes to analyse. We have refined our mRNAs and CUTs datasets to exclude those harbouring a expressed gene within a 500 bp-window downstream of the annotated termination site. This reduced our set of mRNAs to 3393 (out of 5752) and our set of CUTs to 321 (out of 329). Our set of snoRNAs was already reduced in the former versions of the paper to 49 using the same criterion. We have used this new sets to perform metagene analyses using the average instead of the median as the summarizing function with the whole set of mRNAs and CUTs for both the RNAPII and the Sen1 CRAC datasets (see new appendix figure S2 and S5). The profiles obtained are almost identical as the previous ones generated with the median (see new appendix figure S2 and S5). We have also generated the same metagene profiles on quartiles, as suggested, and, although for the data corresponding to the lowest RNAPII and Sen1 signals the profile is a bit more noisy, the general trend is the same for all quartiles (see new appendix figure S2 and S5). For RNAPII CRAC we have also included heatmap analyses with the log₂ fold-change of the different mutants relative to the wt for CUTs and snoRNAs (see new figure EV4). Taken together, the new heatmap and metagene analyses, in addition to the previous metagene analyses and quantifications of each target individually fully support the former conclusions of our manuscript.

4- The question was solved by the novel CRAC experiment.

Thank you for submitting the revised manuscript and addressing the last referee concerns. In a final revision, I would now ask you to take care of several editorial issues that are listed in detail below. Please make any changes to the manuscript text in the attached document only using the "track changes" option. Once these minor issues are resolved, we will be happy to formally accept the manuscript for publication.

YOU MUST COMPLETE ALL CELLS WITH A PINK BACKGROUND ↓
PLEASE NOTE THAT THIS CHECKLIST WILL BE PUBLISHED ALONGSIDE YOUR PAPER

Corresponding Author Name: Odil Porrua
Journal Submitted to: EMBO Journal
Manuscript Number: EMBOJ_2019_101548